# Generalized Correctness Models: Learning Calibrated and Cross-Model Correctness Predictors from Historical Patterns

Hanqi Xiao [1]  Vaidehi Patil [1]  Hyunji Lee [1]  Elias Stengel-Eskin [2]  Mohit Bansal [1]

## Abstract

Generating accurate and calibrated confidence estimates is critical for deploying LLMs in high-stakes or user-facing applications, and remains an open challenge. Prior research has often framed confidence as a problem of eliciting a model's "self-knowledge", i.e., the ability of an LLM to judge whether its own answers are correct; this approach implicitly assumes that there is some privileged information about the answer's correctness that is accessible to the model itself. However, we find that whether trained or training-free, *an LLM attempting to predict the correctness of its own outputs generally performs no better than an unrelated LLM attempting the same task*. Moreover, we hypothesize that a key factor in predicting model correctness, i.e., building a "Correctness Model" (CM), is exposure to a target model's *historical predictions*. We use multiple methods to inject this *historical* correctness information, including training an LLM to predict the confidences of many *other* LLMs, i.e., creating a Generalized Correctness Model (GCM). We use GCMs and CMs as a lens for studying the source of correctness prediction ability and its generalization, studying the importance of answer phrasing, world-knowledge, performance history, in-context examples, and post-hoc calibration for correctness prediction. We evaluate GCMs based on Qwen3-8B across 5 model families and the MMLU, TriviaQA, and Spider datasets, as well as on a downstream selective prediction task, finding that reliable LLM confidence estimation is a cross-model skill learned by encoding correctness history rather than a model-specific skill reliant on introspection. Code: https://github.com/The-Inscrutable-X/CalibratedModelAgnosticCorrectness.

[1]UNC Chapel Hill, USA [2]The University of Texas at Austin, USA. Correspondence to: Hanqi Xiao <hanqix@unc.edu>.

*Proceedings of the 43$^{rd}$ International Conference on Machine Learning*, Seoul, South Korea. PMLR 306, 2026. Copyright 2026 by the author(s).

## 1. Introduction

Confidence information is critical to understanding whether we should trust a system's response to a given query. For Large Language Models (LLMs), confidences enable us to understand honesty in a model (Kadavath et al., 2022), identify hallucinations (Zhou et al., 2025), route to experts when unconfident (Hu et al., 2024), rejection sample (Chuang et al., 2025), and even be leveraged as an RL signal to improve the quality of a model's behavior (Li et al., 2025a). Confidence calibration is the idea that we should enforce a desirable quality for confidences: a calibrated model's confidence should correspond to the empirical rate at which the model's responses are correct, i.e., outputting 90% confidence on an answer should correspond to a 90% chance of the answer being correct; confidence calibration is one desirable quality of confidences, which we analyze alongside performance measures such as accuracy.

Many current approaches to LLM confidence estimation involve asking models to predict the correctness of their own responses, and are rooted in extracting the knowledge that LLMs have about their own correctness (Kadavath et al., 2022; Azaria & Mitchell, 2023; Li et al., 2024; Yin et al., 2023). To measure and improve the calibration of confidence estimates, these approaches also generally inherit frameworks and metrics from forecasting, where it is a standard practice to calibrate forecasts of future events (Degroot & Fienberg, 1983; Guo et al., 2017; Tian et al., 2023). However, a key component is missing in this forecasting analogy: *history*. Human forecasters attempt to calibrate themselves by explicitly recording their confidence on predictions over time and tracking systematic biases, which allows them to adjust and improve their performance (Mellers et al., 2015), albeit imperfectly. Unlike humans – who have privileged information about their own mental states and a memory of their past actions – current LLMs generally approach tasks without an inherent history mechanism for tracking historical performance. Moreover, when framing confidence estimation as a correctness prediction task, it is not clear that any given LLM is better-suited to predict its own correctness. In both cases, given a query $q$, a predicted response $r$ containing a predicted answer $\hat{r}$, the model is simply producing $P_\theta(\text{ is\_correct}(\hat{r}) \mid q, r, \hat{r})$.

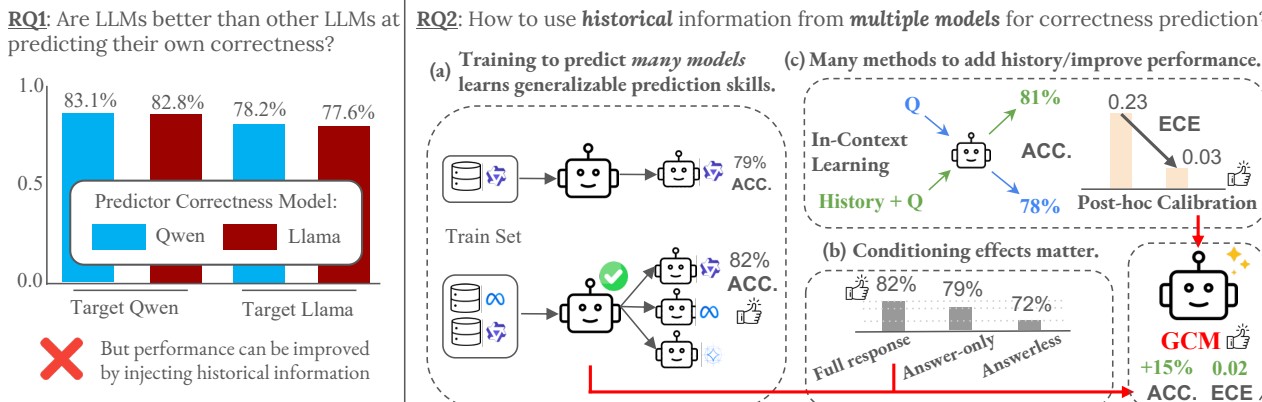

*Figure 1.* **RQ1 & RQ2 overview.** (**Left**) Self- vs. cross-model correctness prediction across Qwen and Llama: accuracies are comparable for each predictor model, suggesting no inherent advantage to a model *predicting its own* outputs. (**Right**) Historical information improves calibration: (a) training on multiple models' histories learns generalizable strategies for correctness prediction; (b) predictive power comes from phrasing of output, CM's world-knowledge, and matching performance to question type. Each stage generalizes, and most prominently strategies for applying world-knowledge; (c) History injected with post-hoc calibration and in-context learning helps improve correctness without finetuning. The GCM combines insights from RQs to achieve high accuracy and extremely low calibration error for the correctness prediction of multiple models, outperforming the logits of equally-sized and larger models.

There is no obvious reason why this prediction should be better when the same LLM parameters $\theta$ were used to produce $r \sim P_\theta(r|q)$. In other words, it remains an open question as to whether models have *self-knowledge*.

We put these assumptions to the test by addressing two core research questions as outlined in Figure 1. First, we ask **RQ1: Are LLMs better than other LLMs at predicting their own correctness?** Our experiments show that for the purposes of obtaining a calibrated confidence score (i.e. a calibrated $P(\text{is\_correct})$), **models have little to no privileged information about their own correctness**. For example, training Llama3-70B to predict its own confidence in being able to answer an MMLU question correctly results in the same performance as training Qwen2.5-72B to do the same, 77.6% vs 78.2% respectively (Figure 1). We observe similar patterns in a training-free setting as well as when providing the answer and question together, indicating that using a model to predict its own confidences offers little to no performance advantage. This allows for the possibility of using one LLM to model the correctness of many others: by removing reliance on self-knowledge, we can improve correctness prediction by learning from the history of many models. Indeed, as demonstrated in Figure 1, we show that which model's history we train on is the clearest predictor for accuracy. Building on these findings, we ask **RQ2: What is the role of historical information from multiple models in calibrated correctness prediction**?

We explore these questions, and subsequent questions that follow from them, by constructing correctness models (CMs), i.e., models designed to provide calibrated $P(\text{is\_correct}(\hat{r})|\cdot)$ scores (which we refer to as $P(c|\cdot)$ as

a shorthand) predicting the correctness of target models (TMs). Unlike prior work, which has generally restricted CMs to the LLM generating responses – either in a zero-shot fashion (Tian et al., 2023) or via finetuning (Kapoor et al., 2024) – or used small linear classifiers (Liu et al., 2024b; Kadavath et al., 2022), **we train LLMs on historical correctness data from multiple different LLMs.** By varying the training data distribution, test settings, post-processing, and input features of the CM, we can concretely test questions and hypotheses about correctness estimation by examining the characteristics of the resulting CM. We investigate the following three axes in RQ2:

1. (RQ2A) **Generalization of CMs trained to predict multiple LLMs**: Do CMs trained on many models' outputs, referred to as Generalized Correctness Models (GCMs), learn generalized strategies for correctness prediction that transfer to other models and datasets? We find that CMs generalize well across different models families and model sizes, even outperforming self-emitted confidences of much larger OOD models, but less well across datasets (Section 3.2).

2. (RQ2B) **Conditioning factors relevant to prediction and generalization**: How do different conditioning variables (e.g., the question $q$, the response $r$, or the predicted answer $\hat{r}$) affect correctness prediction and generalization ability? We measure the incremental gains from adding each variable and find that all components contribute meaningfully; interestingly, answer phrasing plays a substantial role. Moreover, improvements generalize across models, with non-trivial generalization coming from similarities between tasks models find difficult (Section 3.3).

3. (RQ2C) **Alternative methods of encoding history:** Does history incorporated in other ways help improve correctness? We study (a) *post-hoc calibration* and (b) *in-context learning*, which forgoes training in favor of supplying relevant prior examples in-context. We find injecting history via ICL helps improve correctness for larger models, and using post-hoc calibration to map historical confidences to correctness can quickly adapt a CM to dataset-wise OOD settings (Section 3.4).

Our research questions lead to practical insights about developing CMs: RQ2A shows that training Qwen3-8B on the aggregated correctness data from 8 models yields a GCM that outperforms the strongest single-model baseline (directly finetuning eight CMs, one on each target model) by 2.22% accuracy and .041 AUROC on average, observing an improvement on all target models. Moreover, we show that the GCM based on Qwen3-8B outperforms the more powerful Llama3-70B's self-emitted confidences on MMLU by 2.4% absolute accuracy and .265 AUROC. Our GCM also outperforms Qwen3-32B's logit confidences, reducing ECE from .073 to .029 *without having been trained on Qwen3-32B or any other reasoning models*. The GCM transfers across datasets, outperforming a correctness model trained on the target dataset in terms of AUROC, and matching its ECE and accuracy after post-hoc calibration with as little as 5% of the target dataset. Finally, when applied to a downstream task such as selective prediction, we outperform Llama3-70B's logit confidences and a SCM, enabling 30.0%, and 10.8% more coverage at a low 5% risk threshold respectively (See Section 3.5).

## 2. Methods and Experimental Setup

**Correctness Models.** We define a Correctness Model as any system which can provide a confidence that a given query-response pair is correct. This allows us to treat methods such as prompting, probing, auxiliary models, finetuning, and post-hoc calibrators all as parts of correctness models. Mathematically, a Correctness Model is any system that estimates the probability an answer is correct given a query $q$ and a response $r$ containing the answer $\hat{r}$, written as $P(\text{is\_correct}(\hat{r})|q, r, \hat{r})$. For LLMs, the query $q$ is the prompt and the response $r$ is the model's generation given the prompt. For MMLU, we make the distinction that $r$ refers to the model's entire long form response and $\hat{y}$ refers only to the answer choice selected (A,B,C,D).

**Datasets.** Our main analysis is based on the MMLU dataset (Hendrycks et al., 2021) with additional dataset transfer experiments on the TriviaQA dataset (Joshi et al., 2017). To simulate a more realistic setting, we allow models to generate free-form responses and use a judge model with ground truth access to grade them for correctness. Across 8 models in the MMLU dataset, *we elicit long form*

*paragraph length responses averaging 198 tokens*, with responses to math questions often containing reasoning traces that exceed 1000 tokens. A prompt, model response, and binary correctness label of whether the response was correct constitutes a correctness dataset, which is used in this work to inject historical correctness information into CMs. For our main analysis, we build 18 correctness datasets by collecting responses from 10 separate models on the TriviaQA and MMLU datasets (8 from MMLU + 8 from TriviaQA + 2 models on MMLU for OOD testing). We include models from the Gemma3 (Team, 2025a), Qwen2.5 (Qwen et al., 2025), Qwen3 (Team, 2025b), Phi-3 (Microsoft, 2024) and Llama3 families (AIatMeta, 2024), as well as model sizes from 3B to 72B. We additionally evaluate on Spider (Yu et al., 2019), a text-to-SQL semantic parsing benchmark, to test whether the same trends hold outside multiple-choice QA and knowledge-retrieval settings. Further details on datasets are in Section E.

**Measuring Confidence.** Unless otherwise stated, we extract confidences from models via logit based confidences. We note that many uncertainty estimation techniques exist (Kuhn et al., 2023; Geng et al., 2024; Huang et al., 2024) and choose logit based confidences as a common and *extensible* method with strong performance especially after training (we extract logits after training for our main results in Section 3). We elicit **logit based confidences "P(True)"** (Kadavath et al., 2022) by measuring the probability of the token "yes" after exposure to a prompt, model name, and model response appended with the question "Please respond just 'yes' or 'no' in lowercase if the Response correctly answers the Prompt", when ablating the model response, this is rephrased to "Please respond just 'yes' or 'no' in lowercase if [Model Name] will respond correctly to Model Prompt:". Training examples in correctness datasets are structured according to this format with the ground truth yes/no appended. Unless otherwise noted, **all models used in this work are instruction tuned models**.

**RQ1 Setup.** To address RQ1 (Section 3.1), we train two types of Correctness Models with different inputs. We train **Specific Correctness Models (SCMs)** by finetuning a LLM on a correctness dataset to predict the correctness of a response given a query. Excepting for when we explicitly tune these values during ablations, we use LoRA (Hu et al., 2021) with rank 32 and batch size 16 and train for 1 epoch on 70% of a correctness dataset, which is close to 10000 examples for datasets generated from both MMLU and TriviaQA. Unless otherwise noted, we initialize SCMs from a Qwen3-8B model. We utilize a specialized optimal batch size to obtain well calibrated ($\leq .03$ ECE) Correctness Models out of the box with cross-entropy loss (see Section C.4). We train **Answerless Correctness Models** $P(c|q)$ (a more general finetuning-based version of P(IK) from (Kadavath et al., 2022)) by finetuning a LLM to pre-

dict the probability that a target model will respond correctly to a query given only query, without the model response. Answerless CMs can also be seen as assessors (Hernández-Orallo et al., 2022). We use the same hyperparameters as the SCM. We explore verbalized confidences in RQ2 but choose logit confidences for training to avoid approximating soft labels or relying on costly rollout based training. Moreover, logits also allow us to obtain more granular confidence estimates.

**RQ2A Setup.** To analyze the generalization of correctness prediction strategies in RQ2A (Section 3.2) we introduce the General Correctness Model (GCM). We train **General Correctness Models (GCMs)** by finetuning a LLM – in this paper, Qwen3-8B – on the concatenation of 8 correctness datasets under the same training hyperparameters as the Specific Correctness Model. This trains the GCM to predict the correctness of many LLMs. Unlike reward models, we condition on the name of the generating model associated and its performance history. We match the number of training datapoints and training steps between training one GCM to predict 8 LLMs vs training 8 SCMs to predict 8 LLMs, and further ablate impact of training steps in Section C.1. Specifically, we train Qwen3-8B to predict Qwen2.5-3B to 72B, Llama3.1-8B, Qwen3-8B, Gemma3-27B, and Llama3-70B.

**RQ2B Setup.** To explore what parts of a correctness dataset contributes to correctness and what strategies generalize in RQ2B (Section 3.3), we ablate the GCM and SCM into Answerless Correctness Models, and further introduce an Answer-only model type on MMLU as an intermediate ablation. We train **Answer-only Correctness Models** $P(c|q, \hat{r})$ by extracting the answer choice letter from the target model's full response and training a SCM/GCM on the query and answer letter. See Table 1 for probabilistic representations.

*Table 1.* Settings compared in RQ2B, Section 3.3.

| Ablation name | Prob. Form |
| --- | --- |
| Full | $P(c|q, \hat{r}, r)$ |
| Answer-only | $P(c|q, \hat{r})$ |
| Answerless | $P(c|q)$ |

**RQ2C Setup.** We further explore training-free methods in RQ2C (Section 3.4) based on ICL verbalized confidences, and post-hoc calibration. We inject **semantic ICL examples** into models by embedding the train split of a correctness dataset $(q, r, \hat{r}, c)$ into a vector database, and retrieving the top k=5 most semantically similar examples to the current example $(q, r, \hat{r})$ to inject into the prompt for the Correctness Model, we then elicit verbalized confidences. Since we are not focusing on inference efficiency in this setting (ICL prompts have 5x the standard prompt length),

we use verbalized confidences to give a potential performance boost at the cost of efficiency. We elicit **verbalized confidences** (Tian et al., 2023) by prompting the model to give the "calibrated percent probability that the answer will be correct". We **post-hoc calibrate models** by holding out 5% of a correctness dataset and using spline calibration (Lucena, 2018). We also test alternate post-hoc calibration strategies including beta-calibration (Kull et al., 2017), isotonic regression (Zadrozny & Elkan, 2002), or Platt scaling algorithms (Platt, 2000) to map historical probabilities.

**Evaluating Correctness Models.** We evaluate the performance of Correctness Models on 25% of any given correctness dataset, which is close to 3500 examples, ensuring the same questions are used across datasets to prevent train test contamination for GCMs. We highlight a CM's accuracy in predicting correctness as well as their expected calibration error, the standard metrics used for accessing the quality of predicted confidences (Guo et al., 2017). Additionally, due to the variability of metrics like ECE (Guo et al., 2017), we include the Root Mean Squared Calibration Error (Hendrycks et al., 2019) an adaptively binned measurement of calibration. We also include the Area Under the Curve of the Receiver Operating Characteristic (AUROC) which gives a more holistic estimate of predictive power. Importantly, this metric remains sensitive under class imbalance, for example, when a large model such as Gemma3-27b is correct on 78.8% of MMLU questions. We include all setup and prompt details in Section E.

## 3. Results

### 3.1. RQ1: Models have Little Self-Knowledge for Correctness Estimation

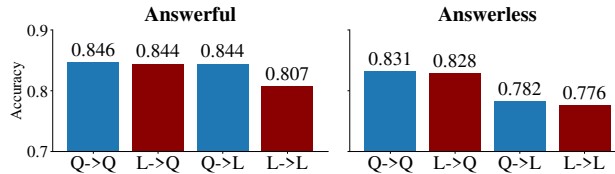

*Figure 2.* **Do LLMs possess special self-knowledge of their correctness?** We compare correctness prediction in *answerful* (with responses) and *answerless* (without responses) settings for Qwen2.5-72B (Q) and Llama3-70B (L). We do not see any advantage for self-prediction (L → L) or (Q → Q). More in Section A.

The motivation for our work comes from the hypothesis that LLMs lack special information about their own correctness. We demonstrate this claim through several experimental settings, highlighting the two most illustrative settings. Given MMLU questions and responses, we first finetune both Qwen2.5-72B and Llama3-70B models to predict each other's correctness as well their own, with the results summarized in the Answerful setting of Figure 2. We find

*Table 2.* Comparing Performance of CMs on MMLU for predicting the correctness of Gemma3-27B and Llama3.1-8B. The General Correctness Model (GCM) outperforms all other baselines in terms of Accuracy and AUROC and achieves low ECE $\leq$ .02.

| Method | Llama3.1-8B | | | | Gemma3-27B | | | |
|---|---|---|---|---|---|---|---|---|
| | Acc | ECE | RMSCE | AUROC | Acc | ECE | RMSCE | AUROC |
| P(True) | .741 | .219 | .253 | .807 | .789 | .197 | .301 | .707 |
| Verbal Confidence | .764 | .160 | .281 | .805 | .797 | .160 | .289 | .738 |
| ICL Verb. Conf. | .743 | .166 | .303 | .785 | .798 | .155 | .302 | .726 |
| Verb. Conf. (Qwen3-32B) | .780 | .161 | .244 | .833 | .807 | .166 | .272 | .725 |
| ICL Verb. Conf. (Qwen3-32B) | .811 | .103 | .186 | .862 | .833 | .119 | .194 | .796 |
| SCM (Trained On Target) | .792 | **.017** | **.069** | .857 | .796 | .037 | .091 | .811 |
| GCM | **.820** | .023 | .080 | .890 | .836 | .029 | .085 | .865 |
| GCM + post-hoc | .818 | .020 | .078 | **.890** | **.836** | **.016** | **.076** | **.865** |

*Table 3.* Comparing GCM and SCM performance on predicting the correctness of a text to SQL task (Spider) for Qwen2.5-7B-Instruct and Llama3.1-8B-Instruct. The GCM improves over the SCM in accuracy and AUROC with similarly low calibration error.

| Method | Qwen2.5-7B | | | | Llama3.1-8B | | | |
|---|---|---|---|---|---|---|---|---|
| | Acc | ECE | RMSCE | AUROC | Acc | ECE | RMSCE | AUROC |
| P(True) | .617 | .339 | .295 | .810 | .595 | .352 | .304 | .891 |
| SCM (Trained On Target) | .766 | **.031** | **.100** | .838 | .873 | .028 | **.089** | .901 |
| GCM | **.805** | .033 | .102 | **.891** | **.898** | **.023** | .090 | **.929** |

that Qwen2.5-72B consistently predicts Llama3-70B's as well as its own correctness better than Llama3-70B does. We attribute this to Qwen2.5-72B being a stronger model with greater parametric knowledge of the true answer to the MMLU questions. (Qwen2.5-72B achieves 83.5% average MMLU accuracy whereas Llama3-70B achieves 78.3%). This shows that using a stronger model is more critical to correctness prediction than "self-knowledge" stemming from using the same model for generation and verification.

To remove the effect of parametric knowledge, we repeat the experiment but remove the model response, so that a greater parametric knowledge will not benefit Qwen2.5-72B. This "answerless" setting allows us to understand whether an LLM can predict the probability it will answer a given question correctly, and evens the playing field between models of differing abilities. In this case we find that Qwen and Llama are roughly equally good at predicting Qwen's correctness, and the same is true when predicting Llama's correctness (shown in Figure 2). If private knowledge existed (such as an internal confidence vector uniquely known to the model itself), we would expect Qwen/Llama to predict their own confidences better. We reinforce these findings by examining training-free settings and additional model families in Section A, finding similar trends suggesting little privileged self-knowledge.

### 3.2. RQ2a: Generalization of CMs Trained to Predict Multiple LLMs

Given that the self-knowledge of the LLM does not provide a noticeable advantage for a Correctness Model, we explore leveraging historical information from multiple models.

**Cross-Model Generalization.** We test whether correctness prediction learned from one model can transfer to others. A Qwen3-8B Generalized Correctness Model (GCM) trained as in Section 2 is evaluated on Llama3.1-8B and Gemma3-27B against Specific Correctness Models (SCMs) trained directly on each. With equal data and training time, the GCM outperforms SCMs by $\geq$3% accuracy on both and achieves $\leq$.03 ECE without post-hoc calibration (Table 2). We replicate the Table 2 setup for TriviaQA in Table 23 and observe similar trends. We further validate the performance of GCM vs SCMs in the Spider text to SQL task in Table 3. In Table 5, we confirm the GCM also outperforms Qwen3-8B trained to predict itself and in Table 4 show the same GCM outperforms Llama3-70B's $P$(True) across all metrics. We next test on models *held out from training*. On Phi-3-mini, the GCM outperforms the SCM by 1.3% accuracy, .009 ECE, and .023 AUROC, while on Qwen3-32B (also held out) it matches the SCM and surpasses Qwen3-32B's zero-shot $P$(True) (Table 6). This suggests that correctness prediction strategies generalizes across families, sizes, and even held-out

*Table 4.* Up Generalization: Qwen3-8B GCM vs. P(True) of Large ID Model (Llama3-70B).

| | ID Large Model (Llama3-70B) | | | |
|---|---|---|---|---|
| Method | Acc | ECE | RMSCE | AUROC |
| P(True) | .798 | .200 | .426 | .584 |
| GCM | **.822** | **.025** | **.078** | **.849** |

*Table 5.* Self Generalization: Qwen3-8B GCM vs. Qwen3-8B trained to predict itself.

| | Self Predict Model (Qwen3-8B) | | | |
|---|---|---|---|---|
| Method | Acc | ECE | RMSCE | AUROC |
| SCM | .814 | .035 | .091 | .835 |
| GCM | **.834** | **.021** | **.071** | **.867** |

*Table 6.* Out-of-Distribution Generalization. Qwen3-8B GCM predicting correctness on Phi-3-mini and Qwen3-32B, models that are held out from the GCM training set.

| | **Phi-3-mini** | | | | **Qwen3-32B** | | | |
|---|---|---|---|---|---|---|---|---|
| **Method** | Acc | ECE | RMSCE | AUROC | Acc | ECE | RMSCE | AUROC |
| P(True) (of target model) | .682 | .042 | .113 | .643 | .870 | .074, | .130 | .861 |
| Specific Model (trained on target) | .787 | .026 | .086 | .853 | **.873** | **.022** | **.072** | .876 |
| General Model (no exposure) | **.800** | **.017** | **.076** | **.876** | .871 | .029 | .084 | **.877** |

*Table 7.* Out-of-Distribution Generalization. GCM trained on MMLU, tested on TriviaQA.

| **Method** | Acc | ECE | RMSCE | AUROC |
|---|---|---|---|---|
| P(True) | .827 | .155 | .277 | .839 |
| SCM (TriviaQA) | .844 | **.023** | **.080** | .895 |
| GCM (MMLU) | .828 | .105 | .150 | .896 |
| GCM + post-hoc | **.844** | .031 | .088 | **.896** |

stronger models. See Section B.3 for GCM vs SCM results on 16 in-distribution correctness datasets.

**Cross-Dataset Generalization.** We test the ability of the generalized model trained on MMLU to predict the correctness of models on TriviaQA in Table 7. We find that although the GCM achieves a similar AUROC to a SCM tuned on TriviaQA and outperforms P(True), it has a lower accuracy and a much higher ECE of .105 compared to the SCM's .023. Surprisingly, this suggests that capabilities generalize better across model families compared to datasets. To establish that GCMs are applicable to settings beyond factual question and answering tasks, we evaluate GCMs on Spider, a semantic parsing benchmark requiring text-to-SQL generation. We show in Table 3 that in the Spider text to SQL task, GCMs also outperform SCMs with trends matching those observed in our MMLU and TriviaQA results (Table 2, Table 23). We study generalizing similarities between models further in Section 3.3. Given the strength of the GCM, outperforming both SCMs and larger models in predicting the correctness of a variety of target models across datasets, we dedicate Section 3.5 to evaluations of the General Correctness Model and its practical applications.

### 3.3. RQ2b: Conditioning Factors Relevant to Prediction and Generalization

**Description of Conditioning Factors Ablated for Analysis.** We successively ablate the query $q$, the answer $\hat{r}$ and the full response $r$ from the correctness dataset to discover impact of each for both the SCM and GCM (Figure 3). We interpret of each ablation as follows: The accuracy gap between $P(c|q, \hat{r}, r)$ (Full) and $P(c|q, \hat{r})$ (Answer-only) ablates the *answer phrasing* of the target model's response, without removing its answer, showing the impact of learning correlations between how the answers are phrased and elaborated with accuracy. This ablation captures, for instance, the difference between seeing "I believe the answer is $4$", and just "$4$"; these findings align with work like Zhou et al. (2024), who study the importance of epistemic markers in confidence, and Stengel-Eskin et al. (2024), who train LLMs to calibrate their use of linguistic signals that communicate confidence. The gap between $P(c|q, \hat{r})$ (Answer-only) and $P(c|q)$ (Answerless) ablates the target model's entire response, but preserves the query, showing the accuracy gain from allowing the CM to leverage its *world-knowledge* to evaluate the likelihood that the answer $\hat{r}$ is correct independent of the past performance of the model on similar questions. Finally, the gap between $P(c|q)$ (Answerless) and $P(c)$ ablates the query, with $P(c|q)$ showing the performance gained by conditioning the target model's past performance on features of the questions compared to a model that simply predicts the majority class; this captures the notion that a given model may differ in its ability to answer different types of questions (Chen et al., 2025).

**Analysis.** We see a substantial increase in accuracy from every ablation, suggesting that every ablated component, including response phrasing, is important to correctness prediction. By additionally comparing the SCM and the

GCM, we find the GCM outperforms SCM by 2% accuracy in the answer-less setting, revealing a correlation between questions LLMs most often answer correctly (indeed, 10 subjects in MMLU see accuracy standard deviations less than 5% across all GCM models). The GCM improved 7% versus the SCM's 4% from answerless to Answer-only, showing that world-knowledge strategies for correctness prediction transfer across models well for questions where a CM can leverage its parametric knowledge (Figure 3). We note that the generalization of world-knowledge shown here and in Section 3.2 requires the CM to leverage its parametric understanding of generated answer vs the true answer, and is similar to that observed in reward models or quality estimators (Lambert et al., 2025; Specia et al., 2009). However, when parametric world-knowledge can not be used (e.g., extremely difficult questions that a CM has little parametric knowledge about, or the answer-less setting), GCMs can still show nontrivial performance (+22% above random, Figure 3) by learning the strengths, output patterns, and weaknesses of various models.

### 3.4. RQ2c: Alternative Methods for Encoding History

We observe in Section 3.1 that stronger models with more parametric knowledge can be better predictors of confidence. Moreover, we note that training the LLM is not always possible, especially with larger LLMs. This motivates us to consider injecting historical information in other ways. We explore two alternative methods: in-context learning (ICL) and post-hoc calibration.

**In-Context Learning.** Rather than training a CM on a dataset of historical examples, we embed the training split of the target model's correctness dataset $(q, r, \hat{r}, c)$ and the current example $(q, r, \hat{r})$, retrieving top k=5 similar training examples to include in-context (details in Section 2). As the ICL setting focuses less on inference efficiency – it requires multiplying prompt length by k – we allowed the model to verbally reason about correctness to further improve accuracy at the cost of inference time. We show in Table 2 that injecting semantically relevant examples from the correctness dataset via ICL improves accuracy by 2.6% and reduces ECE by 4.7% when predicting Gemma3-27B's performance with Qwen3-32B, compared to verbalized confidences without ICL. However, Qwen3-8B shows no gains, suggesting a minimum base capability is needed to benefit.

**Post-hoc Calibration.** Here, we use *post-hoc calibration* to refer to statistical calibration procedures, such as Platt scaling, isotonic regression, beta calibration, and spline calibration, that learn a mapping from model scores or confidences to empirical correctness on held-out historical data. This differs from training-based correctness modeling methods such as SCMs and GCMs, which learn correctness

prediction directly through finetuning. Post-hoc calibration injects historical information by directly aligning an CM's output confidences with the historical ground truth $P(c)$ without conditioning on $q, r$ or $\hat{r}$, as in Section 2. Recall that in RQ2a (Section 3.2), we showed that transfer to new datasets is harder for a GCM: although we outperformed the target SCM in terms of AUROC, the GCM had more than .10 ECE after transfer. However, we find calibrating the result increases accuracy and decreases ECE to match performance of the SCM (Table 7) using only 5% of the target dataset's samples. We additionally observe that it is possible to further calibrate the GCM's output probabilities to reach even lower ECE with post-hoc calibration (Table 2). We use spline calibration for all main results, see Section C.5 for supplementary evaluations of other post-hoc calibrators.

### 3.5. Recommendations for Correctness Prediction

**Building the Most Performant Correctness Model.** Here, we put together the findings from Section 3.2, Section 3.3, and Section 3.4 to summarize the best practices for building a GCM. We recommend the GCM with post-hoc calibration as an accurate and calibrated correctness prediction method. In Table 2 we found that the GCM substantially outperforms strong baselines in-distribution, and transfers without training to beat models trained on OOD target models of different model families, as well as reasoning models Table 6. In addition, when combined with post-hoc calibration, it beats SCMs trained on an OOD target dataset in terms of AUROC, matching it in terms of accuracy and ECE Table 7. Further, the GCM is a inference efficient prefill only method, with one evaluation on MMLU (*3511 examples, ~0.125s/example*) requiring only 7.3 minutes. This solidifies the GCM with Post-hoc calibration as our recommend method of modeling correctness given history. If training is not possible, we recommend using the ICL method presented in Section 3.4. However, we note that while ICL on a significantly stronger model (Qwen3-32B) can match the predictive accuracy of a GCM based on Qwen3-8B, it suffers from high calibration error and has much lower AUROC, which is important for downstream applications such as re-ranking and selective prediction. Additionally, the inference cost of ICL is significantly higher in terms of latency, compute, and memory requirements, due to requiring a large base model, multiplying input prompt length by $k$ for $k$ retrievals, and requiring the generation of a reasoning chain. One MMLU evaluation run (*3511 examples, ~2.6s/example*) already exceeds the cost of training a SCM on correctness.

**Downstream Evaluation on Selective Prediction.** Here, we show that the GCM also provides downstream benefits in a selective prediction task. Selective prediction re-

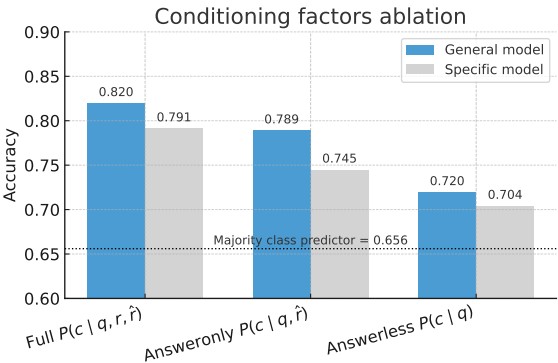

*Figure 3.* GCMs and SCMs across conditioning settings in RQ2b (Section 3.3). More metrics: Table 32.

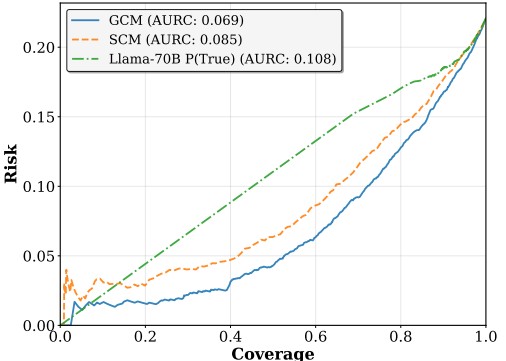

*Figure 4.* **Risk-Coverage Curves** for Selective Prediction, **lower** AURC curves are better.

quires a system to selectively abstain from examples that are unlikely to be correct, with the objective of maximizing coverage (the percentage of examples for which an answer is produced) while minimizing risk (the percentage of predicted answers that were incorrect). Intuitively, the trade-off between coverage and risk is one between usability and safety, with full coverage (no abstention) system having high usability but low safety, while abstaining on all examples (zero coverage) incurs no risk but represents a useless model. Figure 4 shows the risk-coverage curves for the GCM, SCM, and Llama3-70B; here, a lower AURC indicates a better trade-off between coverage and risk. As shown in Figure 4, our results indicate that compared to Llama3-70B's self emitted confidences or a model-specific SCM, the generalized model consistently achieves lower risk (y-axis) at the same level of coverage (x-axis Figure 4). This suggests that the GCM produces more reliable predictions, making it better suited for robust deployment.

## 4. Related Work

**Self-Knowledge and Confidence Calibration.** Calibration, crucial for deciding when to trust AI systems, has been studied in neural models (Naeini et al., 2015; Guo et al., 2017; Ovadia et al., 2019; Wang et al., 2020a) and more recently in LLMs (Mielke et al., 2022; Kadavath et al., 2022; Kuhn et al., 2023; Stengel-Eskin & Van Durme, 2023b; Stengel-Eskin et al., 2024; Tian et al., 2023). Early work showed models like T5, BART, and GPT-2 are poorly calibrated on QA, motivating post-hoc and fine-tuning methods (Jiang et al., 2021). Other studies examined overconfidence in dialogue (Mielke et al., 2022), prompting-based calibration (Kadavath et al., 2022), and fine-tuning (similar to SCMs) with correctness labels (Kapoor et al., 2024). Further efforts probed unanswerable questions (Yin et al., 2023), lying behavior via hidden activations (Azaria & Mitchell, 2023), and black-box elicitation through prompt-

ing, sampling, and aggregation (Xiong et al., 2024). Debate exists as to whether LLMs have privileged access to aspects of their behavior, such as preferences, decode temperature, and detection of activation steering (Binder et al., 2025; Comsa & Shanahan, 2025; Lindsey, 2025; Song et al., 2025). In contrast, we analyze a practical task where introspection can be helpful but is not directly the end goal, proposing a solution to calibrate *multiple* LLMs at once.

**Correctness Models and Cross-Model Transfer.** Another line of work uses *correctness models (CMs)* to predict whether a response is correct. The simplest rely on self-reported confidence (Tian et al., 2023), while stronger methods probe hidden states (Liu et al., 2024b; Kadavath et al., 2022; Beigi et al., 2024; Azaria & Mitchell, 2023; Liu et al., 2024a) or fine-tune LLMs directly on correctness tasks (Kapoor et al., 2024). Surrogate show promise: Shrivastava et al. (2023) report that LLaMA models can outperformance GPT's self-reported confidences, though focused on one-to-one transfer without training. Uncertainty estimation is a related field more closely focused on estimating the model's intrinsic uncertainty rather than correctness (Huang et al., 2024), developing techniques such as semantic uncertainty (Kuhn et al., 2023) to estimate confidence.

**Reward Models, LLM Judges, and Quality Estimation.** While most LLM reward models (RMs) provide pairwise judgements to align with human preference, closest to our work, LLM Judges, Outcome Reward Models, and Process Reward Models are trained to predict the reward of model outputs; when the reward is accuracy-based, these models produce rewards or Likert judgments reflecting answer correctness (Lambert et al., 2025; Tan et al., 2025; Luo et al., 2024; Cobbe et al., 2021; Zheng et al., 2025). Quality estimation (QE) is another line of work have been applied to estimate the quality of a response without ground truth reference, the predominant application focuses on judging the

quality of machine translation outputs (Specia et al., 2009; Zhao et al., 2024). By design, RMs/QE do not consider historical information as they are trained with no knowledge of which model produced the answers they are judging (Lambert et al., 2025; Tan et al., 2025; Zheng et al., 2025). We cover the practical consequences of this difference in Section 3.3 and Section D.

**Assessor Models.** Assessor models are trained to predict the correctness of a model's potential response given only the query (Hernández-Orallo et al., 2022), and are useful for the purposes of routing and preemptive rejection if it is clear that a model has a low chance of responding correctly (Pacchiardi et al., 2025; Zhou et al., 2022; Pacchiardi et al., 2024). Assessors can be seen as an ablation of our method that does not consider responses, which we refer to as Answerless Correctness Models (Section 2). Moreover, they are also similar to the P(IK) setting in the confidence calibration literature (Kadavath et al., 2022). We analyze Answerless Correctness Models / Assessors as part of a hierarchy of conditioning factors for correctness prediction in Section 3.3.

**Downstream Applications.** Correctness estimation has been leveraged to improve downstream tasks. Improved calibration benefits hallucination detection and truthfulness (Zhou et al., 2025; Li et al., 2024; 2025a), enhances interpretability (Stengel-Eskin et al., 2024), strengthens reasoning ability (Wang et al., 2024b; Li et al., 2025a), improves semantic parsing (Stengel-Eskin & Van Durme, 2023a), and supports reliable deployment in system-level routing setups (Hu et al., 2024; Wang et al., 2024a; Ong et al., 2024). Our GCM advances this line of work by providing a cross-model, history-aware framework for correctness estimation that generalizes across both models and datasets. See Section D for additional related works.

## 5. Conclusion

The insight that LLMs have little self-knowledge for the purpose of correctness prediction is counterintuitively beneficial for training Correctness Models. We find that a General Correctness Model (GCM) based on a LLM, trained to predict the correctness of many LLMs, is able to generalize and learn transferable correctness prediction strategies across a variety of models, suffering no penalty for predicting models apart from itself. GCMs outperform both models trained to predict their own correctness and the self-emitted correctness confidences of much larger models. We further show improvements in a downstream selective prediction evaluation stemming from the GCMs' generalization ability.

## Acknowledgments

This work was supported by NSF-AI Engage Institute DRL-2112635, DARPA ECOLE Program No. HR00112390060, NSF-CAREER Award 1846185, and a Capital One Research Award. The views contained in this article are those of the authors and not of the funding agency. We would like to thank Salvador Robles for his help in validating the accuracy of the LLM Judge used for the construction of correctness datasets.

## Impact Statement

This paper presents work whose goal is to advance the field of machine learning. There are many potential societal consequences of our work, none of which we feel must be specifically highlighted here.

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

# A. Extensions of RQ1 Results: Do models have an advantage when predicting their own correctness?

See Tables 8 - 22 for a consolidated comparison of accuracy, calibration (ECE/RMSCE) and AUROC across answer-less, answerful, and untrained settings, covering both within-model and cross-model transfers. We conduct tests across 5 model families and model sizes ranging from 3B to 72B. The training hyperparameters used follow that of Specific Correctness Models (SCMs Section 2). We always conduct nxn tests in order to make relative differences clear and ensure that confounding variables like how lora interacts with a specific architecture/size are not obstacles to spotting privileged self-prediction capability. In our tests, we see no particular advantage to self prediction with any consistency.

Qwen2.5-72B and Llama3-70B have the same number of hidden layers and hidden size (this means the interaction of lora with these two models might be expected to be very similar in terms of optimization and representation power), as such Table 12 - 13 may be of particular interest and show clearly that there is no particular advantage to self-prediction.

*Table 8.* Untrained setting (row-wise). No tuning or epistemic supervision used.

| Configuration | Acc | ECE | RMSCE | AUROC |
|---|---|---|---|---|
| Qwen2.5-7B→Qwen2.5-7B | 0.638 | 0.272 | 0.221 | 0.565 |
| Llama3.1-8B→Qwen2.5-7B | 0.708 | 0.204 | 0.193 | 0.656 |
| Qwen2.5-7B→Llama3.1-8B | 0.552 | 0.331 | 0.242 | 0.520 |
| Llama3.1-8B→Llama3.1-8B | 0.657 | 0.292 | 0.229 | 0.668 |

*Table 9.* Answerless setting (row-wise) with Qwen3-8B and Qwen2.5-7B.

| Configuration | Acc | ECE | RMSCE | AUROC |
|---|---|---|---|---|
| Qwen2.5-7B→Qwen2.5-7B | 0.728 | 0.018 | 0.089 | 0.719 |
| Qwen3-8B→Qwen2.5-7B | 0.737 | 0.029 | 0.092 | 0.751 |
| Qwen2.5-7B→Qwen3-8B | 0.749 | 0.023 | 0.086 | 0.714 |
| Qwen3-8B→Qwen3-8B | 0.765 | 0.036 | 0.094 | 0.756 |

*Table 10.* Answerless setting (row-wise), grouped by *target model*.

| Configuration | Acc | ECE | RMSCE | AUROC |
|---|---|---|---|---|
| Qwen2.5-7B→Qwen2.5-7B | 0.737 | 0.016 | 0.081 | 0.730 |
| Llama3.1-8B→Qwen2.5-7B | 0.731 | 0.024 | 0.086 | 0.737 |
| Qwen2.5-7B→Llama3.1-8B | 0.690 | 0.024 | 0.091 | 0.703 |
| Llama3.1-8B→Llama3.1-8B | 0.694 | 0.016 | 0.084 | 0.728 |

*Table 11.* Answerful setting (row-wise). Models are given access to the predicted answer.

| Configuration | Acc | ECE | RMSCE | AUROC |
|---|---|---|---|---|
| Qwen2.5-7B→Qwen2.5-7B | 0.768 | 0.019 | 0.081 | 0.791 |
| Llama3.1-8B→Qwen2.5-7B | 0.761 | 0.024 | 0.079 | 0.790 |
| Qwen2.5-7B→Llama3.1-8B | 0.751 | 0.022 | 0.078 | 0.804 |
| Llama3.1-8B→Llama3.1-8B | 0.730 | 0.021 | 0.084 | 0.775 |

*Table 12.* Answerless setting (row-wise) with Qwen2.5-72B and Llama3-70B.

| Configuration | Acc | ECE | RMSCE | AUROC |
|---|---|---|---|---|
| Qwen2.5-72B→Qwen2.5-72B | .831 | .029 | .087 | .763 |
| Llama3-70B→Qwen2.5-72B | .828 | .027 | .091 | .732 |
| Qwen2.5-72B→Llama3-70B | .782 | .028 | .080 | .753 |
| Llama3-70B→Llama3-70B | .776 | .028 | .102 | .747 |

One possible concern with the finetuned self- vs. cross-prediction experiments in Section 3.1 is that finetuning may obscure whether LLMs have privileged information about their own correctness. To address this, we evaluate a grid of training-free correctness predictors across five target models and five correctness models. Table 22 reports AUROC.

*Table 13.* Answerful setting (row-wise) with Qwen2.5-72B and Llama3-70B.

| Configuration | Acc | ECE | RMSCE | AUROC |
|---|---|---|---|---|
| Qwen2.5-72B→Qwen2.5-72B | .846 | .025 | .087 | .838 |
| Llama3-70B→Qwen2.5-72B | .844 | .025 | .086 | .784 |
| Qwen2.5-72B→Llama3-70B | .844 | .029 | .087 | .875 |
| Llama3-70B→Llama3-70B | .807 | .029 | .100 | .788 |

*Table 14.* Additional RQ1 Large Models Experiments (Answerless) — Accuracy (Acc)

| Target \ Correctness | Qwen2.5-72B | Llama3-70B | Gemma3-27B |
|---|---|---|---|
| Qwen2.5-72B | 0.831 | 0.828 | 0.827 |
| Llama3-70B | 0.782 | 0.776 | 0.774 |
| Gemma3-27B | 0.784 | 0.784 | 0.791 |

*Table 15.* Additional RQ1 Large Models Experiments (Answerless) — Expected Calibration Error (ECE)

| Target \ Correctness | Qwen2.5-72B | Llama3-70B | Gemma3-27B |
|---|---|---|---|
| Qwen2.5-72B | 0.029 | 0.027 | 0.031 |
| Llama3-70B | 0.028 | 0.028 | 0.037 |
| Gemma3-27B | 0.030 | 0.026 | 0.029 |

*Table 16.* Additional RQ1 Large Models Experiments (Answerful) — Accuracy (Acc)

| Target \ Correctness | Qwen2.5-72B | Llama3-70B | Gemma3-27B |
|---|---|---|---|
| Qwen2.5-72B | 0.846 | 0.844 | 0.843 |
| Llama3-70B | 0.844 | 0.807 | 0.822 |
| Gemma3-27B | 0.849 | 0.817 | 0.825 |

*Table 17.* Additional RQ1 Large Models Experiments (Answerful) — Expected Calibration Error (ECE)

| Target \ Correctness | Qwen2.5-72B | Llama3-70B | Gemma3-27B |
|---|---|---|---|
| Qwen2.5-72B | 0.025 | 0.025 | 0.024 |
| Llama3-70B | 0.029 | 0.029 | 0.016 |
| Gemma3-27B | 0.023 | 0.033 | 0.022 |

*Table 18.* Additional RQ1 Adjacent Gemma Size Experiments (Answerful) — Accuracy (Acc)

| Target \ Correctness | Gemma3-4B | Gemma3-12B | Gemma3-27B |
|---|---|---|---|
| Gemma3-4B | 0.687 | 0.817 | 0.831 |
| Gemma3-12B | 0.772 | 0.796 | 0.829 |
| Gemma3-27B | 0.794 | 0.812 | 0.825 |

*Table 19.* Additional RQ1 Adjacent Gemma Size Experiments (Answerful) — Expected Calibration Error (ECE)

| Target \ Correctness | Gemma3-4B | Gemma3-12B | Gemma3-27B |
|---|---|---|---|
| Gemma3-4B | 0.028 | 0.031 | 0.017 |
| Gemma3-12B | 0.026 | 0.028 | 0.022 |
| Gemma3-27B | 0.025 | 0.025 | 0.022 |

We do not observe a systematic self-prediction advantage in this setting: diagonal entries, where the target model and correctness model are the same, are not consistently higher than off-diagonal entries. Instead, the strongest correctness

*Table 20.* Additional RQ1 Experiments with varied models (Answerless) - Accuracy (Acc)

| Target / Correctness | Mistral-7B | Qwen2.5-7B | Llama3.1-8B | Qwen3-8B |
|---|---|---|---|---|
| Mistral-7B | 0.686 | 0.669 | 0.681 | 0.670 |
| Qwen2.5-7B | 0.731 | 0.728 | 0.726 | 0.738 |
| Llama3.1-8B | 0.691 | 0.681 | 0.677 | 0.702 |
| Qwen3-8B | 0.751 | 0.749 | 0.750 | 0.766 |

*Table 21.* Additional RQ1 Experiments with varied models (Answerless) - ECE

| Target / Correctness | Mistral-7B | Qwen2.5-7B | Llama3.1-8B | Qwen3-8B |
|---|---|---|---|---|
| Mistral-7B | 0.024 | 0.025 | 0.018 | 0.026 |
| Qwen2.5-7B | 0.014 | 0.024 | 0.028 | 0.032 |
| Llama3.1-8B | 0.021 | 0.022 | 0.022 | 0.027 |
| Qwen3-8B | 0.023 | 0.020 | 0.017 | 0.035 |

predictors tend to be stronger models overall, supporting our main claim that, for the purpose of obtaining calibrated correctness estimates, models do not appear to have reliable privileged access to their own correctness.

## B. Extensions of RQ2 Results

### B.1. TriviaQA Results

We replicate Table 2 on the TriviaQA dataset (Joshi et al., 2017) and show the results in Table 23. We find very similar patterns to our MMLU results, where the GCM + Post-hoc Calibration method outperforms SCMs and other methods.

### B.2. Joint Training on MMLU and TriviaQA

In Section 3.2, we observe that GCMs generalize strongly across model families but less cleanly across datasets. To test whether this cross-dataset transfer issue can be mitigated, we train a joint GCM on both MMLU and TriviaQA correctness datasets. We then compare against GCMs trained only on the corresponding target dataset.

As shown in Tables 24 and 25, joint training improves accuracy and AUROC on both datasets for both target models. On MMLU, joint training also improves calibration metrics. On TriviaQA, joint training improves accuracy and AUROC, although ECE and RMSCE are slightly worse than the TriviaQA-only GCM. Overall, these results suggest that training on multiple datasets can improve cross-dataset correctness prediction, but calibration may still benefit from dataset-specific post-hoc calibration.

### B.3. Additional Evaluations for GCMs vs SCMs

We train a batch of 16 SCMs and compare their performances to GCMs on MMLU and TriviaQA. We use a different seed for training (but the same dataset examples) as compared to Table 2 and Table 23. Accuracy impacts are within .2% and patterns stay consistent, showing that GCMs outperform SCMs.

### B.4. MMLU

We compare the General Correctness Model (GCM) against Specific Models (SCMs) on MMLU across 8 models. The GCM outperforms SCMs on all metrics:

- **Accuracy:** +.022 on average (.829 vs. .807), with all 8 cases improving. Largest gain: `Gemma3-27b-it` (+.038). Smallest gain: `Qwen2.5-32B-Instruct` (+.009).

- **ECE:** −.002 on average (.024 vs. .026, lower is better). Improvements: 5 cases; Degradations: 3 cases. Largest decrease (best improvement): `Qwen3-8B` (−.013). Largest increase (worst degradation): `Llama3.1-8B-Instruct` (+.007).

*Table 22.* Training-free self- vs. cross-prediction AUROC across five target models and five correctness models. Diagonal entries correspond to self-prediction. We do not observe a consistent advantage for self-prediction over cross-prediction.

| Target Model \ Correctness Model | Mistral-7B-Instruct | Qwen2.5-7B-Instruct | Llama3.1-8B-Instruct | Qwen3-8B | Gemma3-4b-it |
|---|---|---|---|---|---|
| Mistral-7B-Instruct | .633 | .758 | .742 | .844 | .736 |
| Qwen2.5-7B-Instruct | .553 | .695 | .720 | .823 | .693 |
| Llama3.1-8B-Instruct | .532 | .679 | .717 | .819 | .678 |
| Qwen3-8B | .506 | .710 | .716 | .794 | .633 |
| Gemma3-4b-it | .520 | .727 | .677 | .822 | .587 |

*Table 23.* Comparing Performance of different CMs on TriviaQA for predicting the correctness of Gemma3-27B and Llama3.1-8B. The General Correctness Model (GCM) outperforms all other baselines in terms of Accuracy by 1-4% and achieves extremely low ECE $\leq$ .023.

| Method | Llama3.1-8B | | | | Gemma3-27B | | | |
|---|---|---|---|---|---|---|---|---|
| | Acc | ECE | RMSCE | AUROC | Acc | ECE | RMSCE | AUROC |
| P(True) | .827 | .155 | .277 | .839 | .827 | .164 | .331 | .687 |
| Verbal Confidence | .834 | .136 | .323 | .821 | .825 | .158 | .344 | .687 |
| ICL Verb. Conf. | .827 | .119 | .234 | .855 | .826 | .145 | .254 | .755 |
| Verb. Conf. (Qwen3-32B) | .815 | .151 | .231 | .856 | .831 | .154 | .254 | .747 |
| ICL Verb. Conf. (Qwen3-32B) | .840 | .109 | .202 | .877 | .843 | .128 | .229 | .785 |
| Specific Model | .844 | .023 | .086 | .895 | .839 | .028 | .079 | .843 |
| General Model | .847 | .029 | .090 | .905 | .862 | .028 | .074 | .881 |
| General Model + Post-hoc | **.847** | **.023** | **.077** | **.905** | **.862** | **.018** | **.072** | **.881** |

- **RMSCE:** –.004 on average (.079 vs. .083, lower is better). Improvements: 7 cases; Degradations: 1 case. Largest decrease (best improvement): `Qwen3-8B` (–.014). Largest increase (worst degradation): `Llama3.1-8B-Instruct` (+.010).

- **AUROC:** +.041 on average (.866 vs. .825), with all 8 cases improving. Largest gain: `Qwen2.5-72B-Instruct` (+.064). Smallest gain: `Qwen2.5-7B-Instruct` (+.030).

The largest accuracy gain is observed for `Gemma3-27b-it` (+.038), while the largest AUROC gain is for `Qwen2.5-72B-Instruct` (+.064). Calibration metrics Calibration error stays extremely low <3% for both GCM and SCM, with some improvements observed for GCM. Refer to Table 26 for the full results.

## B.5. TriviaQA

We compare the General Correctness Model (GCM) against Specific Models (SCMs) on TriviaQA across 8 models. The GCM outperforms SCMs on all metrics:

- **Accuracy:** +.021 on average (.865 vs. .844). Improvements: 7 cases; Degradations: 1 case. Largest gain: `Qwen3-8B_supplement` (+.042). Smallest gain: `Llama3.1-8B-Instruct` (+.002). One degradation: `Llama3-70B-Instruct_supplement` (–.005).

- **ECE:** –.0005 on average (.0233 vs. .0238, lower is better). Improvements: 4 cases; Degradations: 4 cases. Largest decrease (best improvement): `Qwen2.5-72B-Instruct` (–.015). Largest increase (worst degradation): `Qwen2.5-3B-Instruct` (+.008).

- **RMSCE:** –.0018 on average (.0775 vs. .0793, lower is better). Improvements: 5 cases; Degradations: 3 cases. Largest decrease (best improvement): `Qwen2.5-72B-Instruct` (–.015). Largest increase (worst degradation): `Llama3-70B-Instruct_supplement` (+.014).

- **AUROC:** +.031 on average (.901 vs. .869). Improvements: all 8 cases. Largest gain: `Qwen2.5-32B-Instruct` (+.047). Smallest gain: `Llama3.1-8B-Instruct` (+.010).

*Table 24.* MMLU results for dataset-specific GCMs versus a joint GCM trained on both MMLU and TriviaQA. Joint training improves accuracy, calibration, and AUROC for both target models.

| Target Model | Accuracy | ECE | RMSCE | AUROC |
|---|---|---|---|---|
| *MMLU GCM* | | | | |
| Llama3.1-8B-Instruct | .820 | .023 | .080 | .890 |
| Gemma3-27b-it | .836 | .029 | .085 | .865 |
| *Joint GCM* | | | | |
| Llama3.1-8B-Instruct | **.827** | **.022** | **.078** | **.898** |
| Gemma3-27b-it | **.840** | **.015** | **.074** | **.871** |

*Table 25.* TriviaQA results for dataset-specific GCMs versus a joint GCM trained on both MMLU and TriviaQA. Joint training improves accuracy and AUROC for both target models, while the TriviaQA-only GCM remains slightly better calibrated.

| Target Model | Accuracy | ECE | RMSCE | AUROC |
|---|---|---|---|---|
| *TriviaQA GCM* | | | | |
| Llama3.1-8B-Instruct | .847 | **.029** | **.090** | .905 |
| Gemma3-27b-it | .862 | **.028** | **.074** | .881 |
| *Joint GCM* | | | | |
| Llama3.1-8B-Instruct | **.849** | .046 | .103 | **.910** |
| Gemma3-27b-it | **.868** | .029 | .084 | **.892** |

The strongest accuracy gain is seen for `Qwen3-8B_supplement` (+.042), while the largest AUROC gain is for `Qwen2.5-32B-Instruct` (+.047). Calibration error stays extremely low <3%, with some improvements observed for GCM. `Qwen2.5-72B-Instruct` shows the largest calibration improvement. Refer to Table 27 for the full results.

## C. Further Discussion and Ablations

### C.1. Unlimited Training Time Ablation

For our main analysis we train the General Model for the same number of epochs as the specific model to match training time. In such a case, training multiple specific models and training one general model would have approximately the same training time cost. We show an ablation here, that even given an unlimited amount of training time (until overfitting occurs), the GCM still outperforms SCMs (Table 28).

### C.2. Figure 1 Experimental Settings

For RQ1, the graph is a copy of the Answerless setting from Figure 2 and described in Section 3.1. For RQ2a, we compare the SCM and GCM when predicting the correctness of Llama3.1-8B, matching the SCM and GCM rows from Table 2 for Llama3.1-8B. For RQ2b, we display the numbers from the conditioning ablations for the GCM, also shown in Figure 3. For RQ2c, we display the gains for predicting Llama3.1-8B using Qwen3-32B with and without ICL. For the GCM, we compare against Llama3.1-8B's logit confidences as defined in Section 2.

### C.3. Discussion on ChatGPT's Memory System and Similar Techniques for Injecting History

We discussed the lack of historical information for LLM based systems in Section 1. We would like to point out that systems such as ChatGPT incorporates a history function. However, we make the distinction that what is necessary is to inject *historical correctness information*, not simply historical information. Additionally, systems such as ChatGPT preserve sparse memories that do not always give a direct account of the performance of their own previous generations, or indeed, even the generations themselves.

### C.4. Discussion on the Effects of Batch Size on Calibration Error

**Minimizing ECE with Training Batch Size.** Prior work has sometimes attributed miscalibration to the use of the cross-entropy loss or suggested that a different loss function should be used to ensure calibrated models after finetuning (Mukhoti et al., 2020; Damani et al., 2025; Li et al., 2025b). For our experimental setting (training SCMs and GCMs), we find that

*Table 26.* Per-model comparison of Specific Models (SCM) vs. General Correctness Model (GCM) on MMLU. For Accuracy, higher is better; for ECE and RMSCE, lower is better; for AUROC, higher is better. Δ is GCM−SCM.

| Model | Accuracy | | ECE | | RMSCE | | AUROC | |
|---|---|---|---|---|---|---|---|---|
| | SCM | GCM (Δ) | SCM | GCM (Δ) | SCM | GCM (Δ) | SCM | GCM (Δ) |
| Llama3.1-8B-Instruct | .791 | .820 (+.029) | .016 | .023 (+.007) | .070 | .080 (+.010) | .857 | .890 (+.033) |
| Llama3-70B-Instruct | .804 | .822 (+.018) | .029 | .025 (−.003) | .086 | .078 (−.008) | .803 | .849 (+.045) |
| Qwen2.5-32B-Instruct | .825 | .834 (+.009) | .036 | .029 (−.007) | .088 | .087 (−.001) | .806 | .844 (+.038) |
| Qwen2.5-3B-Instruct | .790 | .820 (+.030) | .014 | .020 (+.006) | .079 | .073 (−.007) | .864 | .894 (+.031) |
| Qwen2.5-72B-Instruct | .839 | .849 (+.010) | .016 | .014 (−.003) | .072 | .069 (−.003) | .776 | .840 (+.064) |
| Qwen2.5-7B-Instruct | .792 | .816 (+.024) | .026 | .030 (+.004) | .086 | .085 (−.002) | .848 | .878 (+.030) |
| Qwen3-8B | .814 | .834 (+.020) | .035 | .021 (−.013) | .091 | .076 (−.014) | .835 | .867 (+.031) |
| Gemma3-27b-it | .798 | .836 (+.038) | .037 | .029 (−.008) | .094 | .085 (−.009) | .811 | .865 (+.054) |

*Table 27.* Per-model comparison on TriviaQA: Specific Models (SCM) vs. General Correctness Model (GCM). For Accuracy, higher is better; for ECE and RMSCE, lower is better; for AUROC, higher is better. Δ is GCM−SCM.

| Model | Accuracy | | ECE | | RMSCE | | AUROC | |
|---|---|---|---|---|---|---|---|---|
| | SCM | GCM (Δ) | SCM | GCM (Δ) | SCM | GCM (Δ) | SCM | GCM (Δ) |
| Llama3.1-8B-Instruct | .845 | .847 (+.002) | .025 | .029 (+.004) | .082 | .090 (+.007) | .896 | .905 (+.010) |
| Llama3-70B-Instruct. | .890 | .884 (−.005) | .018 | .021 (+.003) | .067 | .080 (+.014) | .800 | .819 (+.019) |
| Qwen2.5-32B-Instruct | .839 | .867 (+.029) | .025 | .022 (−.003) | .080 | .070 (−.011) | .865 | .912 (+.047) |
| Qwen2.5-3B-Instruct | .832 | .872 (+.040) | .015 | .023 (+.008) | .069 | .081 (+.012) | .912 | .944 (+.032) |
| Qwen2.5-72B-Instruct | .875 | .879 (+.003) | .034 | .019 (−.015) | .085 | .070 (−.015) | .861 | .891 (+.030) |
| Qwen2.5-7B-Instruct | .816 | .850 (+.034) | .025 | .023 (−.002) | .086 | .078 (−.008) | .888 | .924 (+.036) |
| Qwen3-8B_supplement | .818 | .860 (+.042) | .020 | .021 (+.002) | .078 | .077 (−.002) | .888 | .928 (+.040) |
| Gemma3-27b-it | .840 | .862 (+.022) | .028 | .028 (−.000) | .086 | .074 (−.012) | .844 | .881 (+.038) |

batch size has a large effect on calibration. We observe that a small batch size of 1 is especially detrimental and higher batch sizes than 32 can also harm ECE. We build our SCMs and GCMs using a batch size of 16 based on this observation (Table 29).

## C.5. Further Discussion on Post-hoc Calibration

For results in the main paper we make use of the spline calibration (Lucena, 2018) post-hoc calibration method. For spline calibration, we use a number of knots equal to 2 or 4 for all results. We explore other calibration methods and their influence on calibration for SCMs of different batch sizes in Table 30. We choose spline calibration after observing positive results for calibration error reduction and that spline calibration can preserve a smooth probability distribution in some cases. For Platt scaling and related methods (Platt, 2000), the output probability distribution is directly scaled, or compressed, this precludes predictions of a high confidence if the distribution is scaled. Spline calibration does not directly compress the distribution and has the possibility of retaining some high probability predictions.

In Section 3.4, we primarily use spline calibration as our post-hoc calibration method following the previous experiments. Here, we additionally compare seven post-hoc calibration methods on TriviaQA following our main the experimental setting, reporting the average improvement across eight target models. Positive values indicate improvements in calibration error, while negative values indicate degradation.

As shown in Table 31, most methods change ECE by less than one percentage point on average. This suggests that GCMs are already relatively well calibrated before post-hoc calibration. Among the tested methods, temperature scaling and spline calibration give small average improvements, while Platt scaling, isotonic regression, ENIR, and BBQ reduce calibration quality in this setting.

*Table 28.* **Unlimited training time ablation**. Columns report Accuracy (avg_correct), Binary ECE (↓), RMSCE (↓), and AUROC (↑). Under conditions where the SCM is allowed to train for as many iterations as necessary until validation loss starts to increase, the GCM is still able to outperform the SCM. Metrics shown for GCM and SCM predicting the correctness of Llama3.1-8B.

| Method | Acc | ECE | RMSCE | AUROC |
|---|---|---|---|---|
| Optimal SCM | .822 | .023 | .073 | .894 |
| Optimal GCM | .845 | .035 | .087 | .912 |

*Table 29.* Uncalibrated accuracy and ECE by `effective-batch-size` for a SCM predicting Qwen2.5-7B.

| effective-batch-size | Acc | ECE |
|---|---|---|
| 128 | .750 | .102 |
| 64 | .780 | .039 |
| 32 | .788 | .030 |
| 16 | .792 | .025 |
| 8 | .795 | .028 |
| 4 | .798 | .066 |
| 2 | .803 | .118 |
| 1 | .810 | .146 |

### C.6. Conditioning Factors Ablations

We include the full metrics for conditioning factors ablations in appendix Table 32.

## D. Expanded Related Work

**Self-Knowledge and Confidence calibration.** Since calibration is essential for deciding when to trust AI systems, prior work has extensively studied calibration in neural models (Naeini et al., 2015; Guo et al., 2017; Ovadia et al., 2019; Wang et al., 2020a), with more recent efforts turning to calibration in large language models (LLMs) (Mielke et al., 2022; Kadavath et al., 2022; Kuhn et al., 2023; Stengel-Eskin et al., 2024; Tian et al., 2023). Early studies found that generative models such as T5, BART, and GPT-2 are often poorly calibrated for QA tasks, requiring post-hoc or fine-tuning methods to better align probabilities with correctness (Jiang et al., 2021). Other works examined overconfidence in dialogue agents and proposed linguistic calibration, matching expressions of doubt with correctness likelihoods, as a remedy (Mielke et al., 2022). Prompting-based methods have also been explored: Kadavath et al. (2022) showed that larger LLMs can produce reasonably calibrated probabilities when asked directly, while Kapoor et al. (2024) argued that prompting alone is insufficient, and that fine-tuning with correctness labels yields better transferable estimates. Additional studies examined unanswerable questions (Yin et al., 2023), lying behavior via hidden activations (Azaria & Mitchell, 2023), and black-box elicitation frameworks combining prompting, sampling, and aggregation (Xiong et al., 2024). Despite these advances, LLMs remain overconfident, and calibration quality improves with scale but falls short of reliability. In contrast to these self-knowledge-based approaches, our work demonstrates that models lack privileged access to their own correctness and introduces a more general solution to calibrate *multiple* LLMs at once.

**Correctness Models and Cross-Model Transfer.** A parallel line of work explicitly uses *correctness models (CMs)* to estimate whether a response is correct. The simplest CMs rely on self-reported confidence from the model itself (Tian et al., 2023), while stronger approaches train linear probes on hidden states (Liu et al., 2024b; Kadavath et al., 2022; Beigi et al., 2024; Azaria & Mitchell, 2023; Liu et al., 2024a) or fine-tune entire LLMs to answer correctness questions directly (Kapoor et al., 2024). Recent studies go beyond surface calibration by modeling *semantic uncertainty*, capturing variability in the meaning of generated outputs, which has been shown to better correlate with correctness (Kuhn et al., 2023). Another intriguing development is the use of surrogate models: Shrivastava et al. (2023) find that even untrained LLaMA models can sometimes predict GPT confidences more accurately than GPT's own self-reported probabilities, suggesting biases in linguistic elicitation. These previous works suggest that correctness signals could potentially transfer across models, but they remain in the one-model-to-one-model setting and do not study the factors that influence the calibration of a correctness model. By contrast, we document these factors and leverage the findings in our Generalized Correctness Model (GCM), which aggregates correctness patterns across many models, providing a more robust and empirically grounded calibration method.

*Table 30.* Calibration results across different effective batch sizes (1, 4, 16). We report Expected Calibration Error (ECE) and Root Mean-Square Calibration Error (RMSCE) for uncalibrated probabilities and post-hoc calibration using Spline (knots=4), Isotonic, Beta, and Platt. Best values (lower is better) are bolded, with ties bolded for both. Results from SCMs trained to predict Qwen2.5-7B.

| Method | batch-size 1 | | batch-size 4 | | batch-size 16 | |
| --- | --- | --- | --- | --- | --- | --- |
| | ECE | RMSCE | ECE | RMSCE | ECE | RMSCE |
| Uncalibrated | .146 | .177 | .066 | .117 | .027 | .085 |
| Spline | .052 | .119 | **.010** | **.074** | **.022** | **.078** |
| Isotonic | **.034** | .140 | .039 | .148 | .032 | .122 |
| Beta | .043 | **.113** | .022 | .078 | **.022** | .084 |
| Platt | .102 | .164 | .055 | .113 | .044 | .097 |

*Table 31.* Average TriviaQA calibration improvement from post-hoc calibration methods across eight target models. Positive values indicate reduced calibration error.

| Method | ECE Improvement Mean | ECE Improvement SE | RMSCE Improvement Mean | RMSCE Improvement SE | N Models |
| --- | --- | --- | --- | --- | --- |
| Spline calibration | +.0023 | .0017 | +.0021 | .0029 | 8 |
| Isotonic regression | −.0053 | .0037 | −.0260 | .0056 | 8 |
| Temperature scaling | +.0027 | .0023 | +.0022 | .0016 | 8 |
| Platt scaling | −.0134 | .0023 | −.0214 | .0029 | 8 |
| Beta calibration | +.0009 | .0020 | +.0001 | .0029 | 8 |
| ENIR | −.0118 | .0026 | −.0383 | .0038 | 8 |
| BBQ | −.0007 | .0022 | −.0256 | .0083 | 8 |

**Reward Models, LLM Judges, and Quality Estimation.** While most LLM reward models (RMs) provide pairwise judgements to align with human preference, closest to our work, LLM Judges, Outcome Reward Models, and Process Reward Models are trained to predict the expected reward of full or partial reasoning outputs; when the reward is accuracy-based, these models produce rewards or Likert judgments reflecting answer correctness (Lambert et al., 2025; Tan et al., 2025; Luo et al., 2024; Cobbe et al., 2021; Zheng et al., 2025). Quality estimation (QE) is another line of work have been applied to estimate the quality of a response without ground truth reference, the predominant application focuses on judging the quality of machine translation outputs (Specia et al., 2009; Zhao et al., 2024). By design, RMs/QE do not consider historical information as they are trained with no knowledge of which model produced the answers they are judging (Lambert et al., 2025; Tan et al., 2025; Zheng et al., 2025). This is necessary to produce unbiased RM/QE models: for example, outputs should not be rewarded just because they come from a model that typically produces correct outputs. In contrast, we focus on learning from "historical patterns", conditioning on the identity of the models generating a response, as well as training on an unaltered history of model performance. Concretely, this allows us to learn both tasks that a specific model is strong/weak in as well as the general difficulty of types of questions for LLMs. Consider any case where the CM/RM/QE has almost no parametric understanding of the true answer (e.g., extremely difficult math), the RM/QE performance would reduce to random as parametric understanding approaches zero (Tan et al., 2025), whereas the Answerless GCM in Figure 3 shows the nontrivial performance of a CM (+22% above random). The work on RM/QE generally does not consider calibrated confidences, RMs output unbounded scalars and QE's ground truth generally lacks probabilistic meaning for calibration.

**Assessor Models.** Assessor models are trained to predict the correctness of a model's potential response given only the query (Hernández-Orallo et al., 2022). Assessor models are useful for the purposes of routing to the most capable model for a given query and preemptive rejection when it is clear that a model has a low chance of responding correctly (Pacchiardi et al., 2025; Zhou et al., 2022; Pacchiardi et al., 2024). Assessors can be seen as an ablation of our method that does not consider responses, which we refer to as Answerless Correctness Models (Section 2). Moreover, they are also similar to the P(IK) setting in the confidence calibration literature (Kadavath et al., 2022). We analyze Answerless Correctness Models / Assessors as part of a hierarchy of conditioning factors for correctness prediction in Section 3.3.

**Downstream Applications.** Correctness estimation has been leveraged to improve downstream tasks. Improved calibration benefits hallucination detection and truthfulness (Zhou et al., 2025; Li et al., 2024; 2025a), enhances interpretability (Stengel-Eskin et al., 2024), strengthens reasoning ability (Wang et al., 2024b; Li et al., 2025a), improves semantic parsing (Stengel-Eskin & Van Durme, 2023a), and supports reliable deployment in system-level routing setups (Hu et al., 2024; Wang et al., 2024a; Ong et al., 2024). Our GCM advances this line of work by providing a cross-model, history-aware

*Table 32.* **Conditioning factors** ablations (Section 3.3), full results on all metrics.

| | Specific Model | | | | General Model | | | |
|---|---|---|---|---|---|---|---|---|
| **Setting** | Acc | ECE | RMSCE | AUROC | Acc | ECE | RMSCE | AUROC |
| Full $P(c \mid q, r, \hat{r})$ | .792 | .017 | .069 | .857 | .820 | .023 | .080 | .890 |
| Answer-only $P(c \mid q, \hat{r})$ | .745 | .023 | .087 | .810 | .789 | .034 | .088 | .852 |
| Answerless $P(c \mid q)$ | .704 | .030 | .101 | .735 | .720 | .024 | .095 | .781 |

framework for correctness estimation that generalizes across both models and datasets.

# E. Additional Details on Experimental Setup

**Further Training Details.** Expanding on Section 2, except for when we explicitly tune these values during ablations, for all GCM and SCM training runs we use LoRA (Hu et al., 2021) with rank 32, batch size 16, alpha 16, dropout 0.0, learning rate 1e-5, targeting default query and value matrices. We mask the loss during finetuning except for the token (yes/no) from which we obtain the logit based confidences "P(True)" introduced in Section 2 and finetune to match the LLM's prediction with the ground truth correctness (yes/no) marker using the standard cross-entropy loss. We used accelerate (Gugger et al., 2022) and trained using DeepSpeed with ZeRO stage 2 (Rajbhandari et al., 2019), mixed precision in `bf16`.

**Correctness Dataset Details.** Section 2 mentions the creation of several correctness datasets for the training and evaluation of CMs. We list all datasets used in this work in Table 33 including the name of the base dataset used to gather questions, and the name of the model used to generate responses. The training data for the GCM is a concatenation of the correctness datasets in block 1 or block 3 depending on whether the GCM was trained on MMLU or TriviaQA, while block 2 are datasets generated for the purposes of OOD evaluations. SCMs were trained on each of the listed datasets. The average length of a MMLU dataset (used for training GCMs) was 198 tokens. The average length of a response from the TriviaQA dataset was 70.4 tokens.

**Setup for Additional Spider Text to SQL Evaluations.** To test whether our findings extend beyond multiple-choice QA and knowledge-retrieval settings, we evaluate correctness prediction on Spider (Yu et al., 2019), a text-to-SQL semantic parsing benchmark. In Spider, the target model must generate a SQL query given a natural-language question and a database schema, and we construct correctness labels using exact-match evaluation. We follow the same setup as our Table 2 experiments, except that the Spider GCM is trained to predict six target models rather than eight. We omit the 70B-scale models and train using the 32B-scale models for time. We use the Spider training split to construct correctness datasets with a 75% train and 25% test split, evaluating generated SQL queries with the exact-match criteria to obtain correctness labels.

**ICL Retrieval Details.** In order to facilitate semantic retrieval for the **semantic ICL examples** setting (Section 2), we utilize the chroma library (Chroma, 2025), and we use the default embed function, which at time of writing is "all-MiniLM-L6-v2" based on (Wang et al., 2020b).

## E.1. Prompt Templates

We document the prompt templates generated under different argument settings in the following paragraphs.

**SCM / GCM / Answer-only / P(True)**

```
You are grading [model name]'s responses to prompts for correctness.
###Model Prompt
{input_prompt}
###Model Response
{cleaned_model_completion}
###Instruction
Please respond just 'yes' or 'no' in lowercase if Model Response
```

*Table 33.* Constructed datasets from different base datasets and generator models.

| Base Dataset | Model Used |
|---|---|
| MMLU | Gemma3-27B-IT |
| MMLU | Llama3.1-8B-Instruct |
| MMLU | Llama3-70B-Instruct |
| MMLU | Qwen2.5-3B-Instruct |
| MMLU | Qwen2.5-7B-Instruct |
| MMLU | Qwen2.5-32B-Instruct |
| MMLU | Qwen2.5-72B-Instruct |
| MMLU | Qwen3-8B |
| MMLU | Qwen3-32B |
| MMLU | Phi-3-mini-4k-instruct |
| TriviaQA | Gemma3-27B-IT |
| TriviaQA | Llama3.1-8B-Instruct |
| TriviaQA | Llama3-70B-Instruct |
| TriviaQA | Qwen2.5-3B-Instruct |
| TriviaQA | Qwen2.5-7B-Instruct |
| TriviaQA | Qwen2.5-32B-Instruct |
| TriviaQA | Qwen2.5-72B-Instruct |
| TriviaQA | Qwen3-8B |

```
correctly answers Model Prompt: {ground truth yes/no}
```

(Note: the trailing "yes" or "no" is inserted according to ground truth `"is_correct"` of the response when used as a training example, when used during inference, it is removed.)

**Answerless Setting**

```
You are predicting the likelihood that [model name] will respond
correctly to a prompt.
###Model Prompt
{input_prompt}
###Instruction
Please respond just 'yes' or 'no' in lowercase if [model name] will
respond correctly to Model Prompt: {ground truth yes/no}
```

(Note: the trailing "yes" or "no" is inserted according to ground truth `"is_correct"` of the response when used as a training example, when used during inference, it is removed.)

**Verbalized Confidence**

```
You are examining the correctness of [model name]'s predictions.
You are given a question and answer pair and you must output the
probability that [model name] has answered the question correctly.
You should make a concise and informative analysis of the current
question and response to determine whether the response has
correctly answered the question.
After your analyses, please respond with a calibrated percent
probability that the answer will be correct in the format
ANSWER_CORRECT_PROBABILITY: xx.xx%

##Current Model Prompt
{input_prompt}
##Current Model Response
{cleaned_model_completion}
```

```
##Please respond with with a verbalized and calibrated percent
probability that the Current Model Response is a correct response
to Current Model Prompt and output your answer in the EXACT format
'ANSWER_CORRECT_PROBABILITY: xx.xx%'.
Be sure to follow the format exactly.
```

**ICL Verbalized Confidence**

```
You are examining the correctness of [model name]'s predictions.
You are given {N} training datapoints consisting of questions
[model name] has been asked in the past.
Training datapoints contain a question, [model name]'s response,
and human labeled yes/no of whether the response was correct.
After the training datapoints you are given the current question
and answer pair and you must output the probability that
[model name] has answered the question correctly.
You should make a concise and informative analysis of the current
question and response to determine whether the response has
correctly answered the question.
Then, if you are still unsure of your decision, you can explicitly
analyze the model's past performance on similar examples and make
appropriate adjustments depending on the relevance of the training
examples.
After your analyses, please respond with a calibrated percent
probability that the answer will be correct in the format
ANSWER_CORRECT_PROBABILITY: xx.xx%

##Previous Performances
Example 0 -- Distance: {d_0} (lower = more similar)
{document_0}

...
Example N -- Distance: {d_N} (lower = more similar)
{document_N}

##Current Model Prompt
{input_prompt}
##Current Model Response
{cleaned_model_completion}
##Please respond with with a verbalized and calibrated percent
probability that the Current Model Response is a correct response
to Current Model Prompt and output your answer in the EXACT format
'ANSWER_CORRECT_PROBABILITY: xx.xx%'.
Be sure to follow the format exactly.
```

(The ICL Verbalized Confidence prompt is not used for training, and thus does not include ground truth labels, except in the included ICL examples, during any form of its usage.)

**Notes for Reproducibility**

- Variable placeholders ({input_prompt}, {cleaned_model_completion}, {document_i}, {d_i}, {ground truth yes/no}) are filled dynamically from the correctness dataset.

- {document_i} and {d_i} refer to results from ICL retrieval described under the ICL Retrieval Details heading, they represent a full training example text with labels formatted according to the SCM/GCM/Answer-only/P(True) format

and the rank of its similarity to the current example under consideration respectively.

- {input_prompt} and {cleaned_model_completion} refer to the MMLU/TriviaQA prompt to the model, and the target model's response.

- For the Answer-only prompt in MMLU, the {cleaned_model_completion} is truncated to only display the answer choice letter.

