# OpenReview forum: "Generalized Correctness Models: Learning Calibrated and Cross-Model Correctness Predictors from Historical Patterns"
_ICML.cc/2026/Conference — ICML 2026 regular_

### Official Review · Reviewer_sZtR · 2026-03-08

**Soundness:** 4
**Presentation:** 4
**Significance:** 1
**Originality:** 2
**Overall Recommendation:** 4
**Confidence:** 2

**Summary:**

The paper questions the assumption that LLMs possess special self-knowledge about the correctness of their own answers. Instead, it proposes Correctness Models (CMs) and Generalized Correctness Models (GCMs) that learn to predict answer correctness using historical performance data from multiple models. Experiments on MMLU and TriviaQA show that models trained on aggregated correctness histories outperform both self-confidence estimates and model-specific predictors.

**Compliance With Llm Reviewing Policy:**

Affirmed.

**Final Justification:**

This work is solid, especially with the additional experiments in the rebuttal. However, from my perspective, it does not address a sufficiently significant topic from the beginning, and I do not see strong practical value.

**Key Questions For Authors:**

1. I am not very familiar with the literature on Correctness Models. As stated in the introduction, potential applications include assessing model honesty, detecting hallucinations, routing queries to expert models when confidence is low, rejection sampling, and serving as a reward model. However, these applications typically arise in real-world interactive settings rather than in tasks like MMLU and TriviaQA (it seems to make more sense to use free-text question answering tasks, e.g., benchmarks like HumanEval). What is the practical value of your proposed approach, and how would your methods translate to real-world deployment scenarios?

**Limitations:**

yes

**Strengths And Weaknesses:**

Strengths:
1. The paper is well-written.
2. The experiments are well-designed.

Weaknesses:
1. The experiment scale is a concern, with only 2 benchmarks included.
2. It would be helpful to include more comparisons with prior work on practical tasks, such as [1]. In particular, Specific Correctness Models may have efficiency advantages, as they can reuse the KV cache to generate confidence scores during inference, potentially reducing computational overhead. A more detailed evaluation of both accuracy and inference efficiency would clarify the trade-offs between different approaches and help identify the practical scenarios in which your proposed method provides the greatest benefit.
3. Unclear practical value, see question 1.

[1] Kang, Zhewei, Xuandong Zhao, and Dawn Song. "Scalable best-of-n selection for large language models via self-certainty." arXiv preprint arXiv:2502.18581 (2025).

---

> ### Author Rebuttal · Authors · 2026-03-31
>
> Thank you for highlighting that our paper is “**well-written**” and that our experiments are “**well-designed**”. We address your questions below.
>
> ### **W1. Experiment scale and practical tasks - Training on Spider Text to SQL**
>
> Thanks for pointing out the need for more realistic settings. In order to show that our methods work beyond MMLU, and further beyond the knowledge retrieval / QA setting, we conduct tests on Spider [1]. In Spider, the model must construct SQL queries in response to questions and a database schema. **We show that GCMs outperforms SCMs on the Spider task as well.** Unfortunately, due to limited time we were unable to set up the environment for Human Eval, but consider Spider to be a similar more realistic task in spirit.
>
> Spider Experiment Results:
>
> | | Accuracy | ECE | RMSCE | AUROC |
> |:-----------------------------------------------|-----------:|------:|--------:|--------:|
> | Untrained Predicts Qwen2.5-7B-Instruct | 0.617 | 0.339 | 0.295 | 0.81 |
> | SCM Predicts Qwen2.5-7B-Instruct | 0.766 | **0.031** | **0.1** | 0.838 |
> | GCM Predicts Qwen2.5-7B-Instruct |      **0.805** | 0.033 |   0.102 |   **0.891** |
>
> | | Accuracy | ECE | RMSCE | AUROC |
> |:-----------------------------------------------|-----------:|------:|--------:|--------:|
> | Untrained Predict Llama-3.1-8B-Instruct | 0.595 | 0.352 | 0.304 | 0.891 |
> | SCM Predict Llama-3.1-8B-Instruct | 0.873 | 0.028 | **0.089** | 0.901 |
> | GCM Predict Llama-3.1-8B-Instruct |      **0.898** | **0.023** |   0.09  |   **0.929** |
>
> Exp Detail: The setup for the experiments are the same as for our table 1 experiments, except for that we train the GCM to predict 6 models (rather than 8), skipping the 70B scale levels and training on the 32B models for time. We use the train set of Spider under the exact match evaluation to construct the correctness dataset. We will report results for all 6 models in the main paper or our next reply (highlighting two models due to word count).
>
> We also note that rather than using the multiple-choice or short answer versions of the MMLU and TriviaQA tasks, **we take questions from the MMLU/TriviaQA datasets and convert them into free form versions** e.g., the average response for MMLU was paragraph length (198 tokens), with many much longer responses, lining up with your observations that ‘it seems to make more sense to use free-text question answering tasks’. We grade the accuracy of the LLM’s responses via a LLM Judge provided with the ground truth answer.
>
> ### **W2. Efficiency of GCMs vs SCMs**
> While it’s possible that certain types of SCMs could reuse the KV cache, we argue that the cost of a GCM is only slightly higher: as we state in section 3.5 (line 265): being a prefill-only method, the cost of a GCM is only 1 more forward pass than generating the sequence of tokens. The major bottleneck LLM speed by far is sequential generation, requiring multiple forward passes. Additionally, while it may be possible for SCMs to reuse the KV cache, this is only true when the SCM trained is based on the model we wish to predict, this is not true in many cases, e.g., with a closed source model, or when predicting correctness for a small model that lacks much parametric knowledge.
>
> We further compare the inference efficiency of our method with baselines such as verbalized confidences (lines 381-383), showing that GCMs are much faster and more efficient.
>
>
> ### **W3 & Q1. Practical Applications**
>
> We show that for tasks where the model is allowed to make a free form response, it is possible to use a GCM e.g., for prediction selection, as shown in Figure 4. Selective prediction allows us to use a model to answer more questions rather than redirecting to human experts while keeping the risk of being incorrect at the same level. For example in Figure 4, **if we fix our risk tolerance at 5%, we can answer around \~30% (absolute) more questions compared to the P(True) of a Llama3-70B base model by using a Qwen3-8B GCM**. This is also an improvement of \~12% (absolute) on top of using a SCM.
>
> Taking Spider or similar semantic parsing / program writing tasks as an example, the user could ask the computer to generate a program, and then use a GCM to check the confidence. If the GCM is unconfident in the correctness of a result, the user can be alerted to check the result or regenerate the result.
>
> We hope our response has addressed all of your questions and will allow you to revisit your score.
>
> [1] Yu, Tao, et al. "Spider: A large-scale human-labeled dataset for complex and cross-domain semantic parsing and text-to-sql task." Emnlp, 2018.

---

> > ### Author Rebuttal · Reviewer_sZtR · 2026-03-31
> >
> > Thank you for your thoughtful responses. They clarify my questions/concerns. I will increase my score to 4.

---

> > > ### Author Response · Authors · 2026-04-07
> > >
> > > Thank you for your engagement and for increasing your score, we are glad that our responses clarified / answered your questions.

---

### Official Review · Reviewer_uuTW · 2026-03-11

**Soundness:** 3
**Presentation:** 3
**Significance:** 3
**Originality:** 3
**Overall Recommendation:** 4
**Confidence:** 3

**Summary:**

The authors challenge the long-standing assumption that LLMs possess privileged "self-knowledge" of their own correctness. To address this, the paper shows that an LLM attempting to evaluate its own outputs generally performs no better than an entirely unrelated LLM evaluating those same outputs. Consequently, the paper introduces the Generalized Correctness Model that is trained on historical predictions of different LLMs with the goal of learning universal, model-agnostic patterns of correctness. Interestingly, the model also generalizes to unseen model families, outperforming self-generated confidence scores of larger target models. Finally, the ablation studies show that the model predicts correctness by analyzing answer phrasing and by cross-referencing the generated answer against its own parametric knowledge.

**Compliance With Llm Reviewing Policy:**

Affirmed.

**Final Justification:**

The authors submitted a detailed response, and I would like to support the acceptance of the paper.

**Key Questions For Authors:**

Please refer to the weaknesses section for some open questions concerning the work.

**Limitations:**

Please refer to the weakness section for limitations.

**Strengths And Weaknesses:**

**Strengths**

1. The empirical analysis in the paper effectively shows that LLM evaluating its own output performs no better than an external, unrelated LLM doing the same task.

2. Figure 1 is very well made and detailed.

3. I liked the ablation experiments that effectively isolate how confidence prediction actually works and showing that they heavily rely on the linguistic artifacts and evaluator's parametric knowledge.

**Weaknesses and Open Questions**

1. While, in Section 3.3, the authors show that answer phrasing is a driver of the model's predictive power, could the proposed approach be vulnerable to how frontier models are trained using RLHF? For instance, most frontier models heavily train their models using RLHF (causing the model to be sycophantic) to generate highly confident phrasing. Will the model's focus on linguistic mechanism for detecting errors be effected due to this behavior?

2. While the correctness model transfers across different model architectures, it underperforms on new domain shifts. For instance, a model trained on the multi-choice MMLU benchmark experiences a massive spike in calibration error when attempting to predict correctness on the short-form TriviaQA dataset. It would be great if the authors could explain and elaborate this finding.

3. Existing work [1] quantify the uncertainty of generated reasoning in LLMs and show that LLMs show 100% confidence when generating chain of thought reasoning. It would be great if the authors can argue about similar works in reasoning with their arguments of LLM confidence when eliciting responses.

[1] Tanneru et al. Quantifying uncertainty in natural language explanations of large language models. AISTATS, 2024.

---

> ### Author Rebuttal · Authors · 2026-03-31
>
> Thank you for acknowledging that our paper “**challenge[s] the long-standing assumption that LLMs possess privileged ‘self-knowledge’ of their own correctness**” and “**effectively shows that LLM evaluating its own output performs no better than an external, unrelated LLM doing the same task.**”
>
> ### **W1. Will the model's focus on linguistic mechanisms for detecting errors be affected by RLHF?**
> Yes, while RLHF can affect the answer's phrasing, we do not require that the phrasing sounds confident/unconfident, but only that there is variation between answer phrasings. The subtle variation in answering phrasing is then learned by the GCM. To model a more realistic setting, we don’t evaluate base models but focus on instruct models (e.g., Qwen2.5-7B) that have been RLHF-ed and still observe this pattern.
> ### **W2. While cross model architecture transfer is high, cross dataset transfer is low?**
> We attribute the cross-dataset transfer performance to the qualitative differences between MMLU and TriviaQA. For TriviaQA, most answers are trivia questions, which may require different strategies for leverage parametric knowledge compared to MMLU (which is more admissible to reasoning and contains math, logic, and computer science questions). Further, the format is not multiple choice, and the length of the responses are much different, for TriviaQA, the length of the responses are typically much shorter, which means patterns from answer formatting will also be different. Because the topics covered are fairly different, it is also hard for the model to leverage much of its knowledge about which models perform well on which types of tasks. Thus following our analysis of various components of correctness prediction ability, **TriviaQA and MMLU are non-trivially OOD with respect to each other**, even though on the surface, they seem to be both Q&A tasks.
> We further train a GCM on both datasets to analyze if it would improve transfer and show some transfer between tasks, especially observing improvements for MMLU prediction on all metrics but less positive effects for TriviaQA, however, we note that the calibration spike is largely mitigated (see RMSCE). The assumption of training on both datasets also admits holding out a small calibration portion (only 5%) to decrease calibration error further, as in Table 6.
>
> MMLU Results
>
> | Target Model               |   Accuracy |   ECE |   RMSCE |   AUROC |
> |:--|--:|--:|--:|--:|
> |MMLU GCM ↓||||
> | Llama-3.1-8B-Instruct     |      0.820 | 0.023 |   0.080 |   0.890 |
> | gemma-3-27b-it            |      0.836 | 0.029 |   0.085 |   0.865 |
> |Joint GCM ↓|||||
> | Llama-3.1-8B-Instruct     | **0.827** | **0.022** | **0.078** | **0.898** |
> | gemma-3-27b-it            | **0.84**  | **0.015** |  **0.074** | **0.871** |
>
> TriviaQA Results
>
> | Target Model               |   Accuracy |   ECE |   RMSCE |   AUROC |
> |:--|--:|--:|--:|--:|
> |TriviaQA GCM ↓||||
> | Llama-3.1-8B-Instruct     |      0.847 | **0.029** |  **0.090** |   0.905  |
> | gemma-3-27b-it            |      0.862 | **0.028** |   **0.074** |   0.881 |
> |Joint GCM ↓|||||
> | Llama-3.1-8B-Instruct     |      **0.849** | 0.046 |   0.103 |   **0.91**  |
> | gemma-3-27b-it            |      **0.868** | 0.029 |   0.084 |   **0.892** |
>
> ### **W3. LLMs show 100% confidence when generating chain of thought reasoning?**
> Thanks for bringing up the fact that models often give 100% confidence to their own reasoning. The paper you link analyzes the verbalized confidences of language models after reasoning [1], we will discuss this as one reason why you should prefer to use a trained logit based classifier like a GCM rather than verbalized confidences from a large LLM. We motivate comparing against the verbalized baseline following the observation of [2] that LLMs have surprisingly improved verbalized confidences as size increases and which are often better than their logit confidences. **We note the statement that LLMs show 100% confidence in their reasoning by [1] applies to directly verbalized confidences, not the linguistic patterns in the answers.** In contrast, GCM does not depend on the verbalized confidences of the LLM, and benefits if there are any linguistic patterns in the reasoning provided that have some correlation with confidence (we show that this is the case in our Figure 3 ablations between Full and Answeronly).
>
> We hope our response has addressed all of your questions and will allow you to revisit your score.
>
> [1] Tanneru et al. Quantifying uncertainty in natural language explanations of large language models. AISTATS, 2024.
> [2] Tian et al. Just Ask for Calibration: Strategies for Eliciting Calibrated Confidence Scores from Language Models Fine-Tuned with Human Feedback. Emnlp, 2023.

---

> > ### Author Rebuttal · Reviewer_uuTW · 2026-04-02
> >
> > Thank you for your detailed rebuttal. Please see below for some remaining concerns.
> >
> > **Re: we focus on instruct models (e.g., Qwen2.5-7B) that have been RLHF-ed and still observe this pattern.**
> >
> > Doesn't this limit the applicability and efficacy of the findings and proposed technique?

---

> > > ### Author Response · Authors · 2026-04-06
> > >
> > > We perform a new experiment to test the applicability of GCMs on a non-RLHF Model. We generate a correctness dataset by rolling out the non-RLHF Qwen2.5-3B model and evaluate a GCM trained only on RLHF-ed models (MMLU, TriviaQA) on predicting the non-RLHF model, showing transfer, suggesting that correctness prediction ability generalizes to non-RLHF-ed Models.
> > >
> > > Results on non-RLHF model
> > > | Correctness Model | Accuracy | ECE | RMSCE | AUROC |
> > > |:----------------|---------:|----:|------:|------:|
> > > | P(True) | 0.437197 | 0.439597 | 0.294573 | 0.856303 |
> > > | GCM (Trained on RLHF only) | **0.702079** | **0.107234** | **0.153725** | **0.852095** |
> > >
> > > **Responses from a non-RLHF model can also be collected into a correctness dataset, meaning that GCMs and SCMs can be trained to predict correctness.**
> > > We do not expect that non-RLHF models will confound any of the three attributes benefiting the generalization of GCMs (answer formatting: usable as linguistic variance still exists in outputs, parametric knowledge: usable as answers must still carry semantic meaning by definition, past performance history: usable as the past performance of non-RLHF models can be tracked as well).
> > >
> > > Additional notes if useful:
> > > We saw that your main concerns involve how RLHF will influence correctness information gained from analyzing answer formatting. So we wanted to briefly note that answer phrasing is only one of the aspects used by correctness models in order to predict correctness and the other aspects used by GCMs should be unaffected even if non-RLHF models for any reason confound the ability of GCMs to use answer formatting information to predict correctness (The answer only model in Figure 3 can be thought of as the lower bound performance in such cases, and continues to outperform SCMs). We chose to focus on instruction tuned and RLHF-ed models mainly because instruction tuned RLHF-ed models are more common and prevalent especially for question answer tasks, and other instruction following tasks such as program writing or semantic parsing.
> > >
> > > Thanks for your detailed reviews and followups, and we hope this response addresses your remaining questions and would allow you to improve your score!

---

### Official Review · Reviewer_oJfq · 2026-03-13

**Soundness:** 2
**Presentation:** 3
**Significance:** 2
**Originality:** 2
**Overall Recommendation:** 3
**Confidence:** 4

**Summary:**

This paper introduces a novel perspective: the models have little to no privileged information about their own correctness. Based on its theory, they train LLMs on historical correctness data from multiple different LLMs as Generalized Correctness Models (GCMs), enabling a single LLM to model the correctness of many other LLMs. Various experiments on different models demonstrate that the GCM is a promising way for various model correction predication. Besides that, this paper also investigates how the different components in the response affect the correctness predication and the effectiveness of different methods to inject historical information.

**Compliance With Llm Reviewing Policy:**

Affirmed.

**Final Justification:**

After considering both the original submission and the authors' rebuttal, I maintain my original recommendation of Weak Reject.

The paper has clear merits. It proposes an interesting perspective on correctness prediction based on historical information, introduces the GCM framework for cross-model correctness modeling, and includes several useful empirical analyses on response components and history encoding methods. The rebuttal also improves the empirical completeness of the paper by adding untuned comparisons, a model-name ablation, stronger post-hoc calibration baselines, and an additional Spider experiment beyond the original QA settings.

However, two central concerns remain only partially resolved. First, the paper's interpretation of the model-name effect is still not fully supported. The new ablation on including versus removing model names shows that historical/model identity information is useful for prediction, but it does not demonstrate that model names act as a trigger for self-specific privileged information. This matters because the paper's contribution is not only an empirical predictor, but also a conceptual claim about why correctness prediction works and why self-knowledge is limited.

Second, the scope of the generalization claim remains unclear. The new Spider result is useful evidence of cross-task transfer and strengthens the empirical story beyond QA. However, it does not address the stronger OOD question I raised: whether, in a genuinely out-of-domain setting, a GCM based on historical information remains competitive with a correctness model trained for that target domain. The rebuttal mainly compares GCM against base logits, so it does not fully support the broader claim that historical-information-based correctness modeling generalizes as a preferable paradigm in new domains.

Overall, the rebuttal improves the paper and provides enough support for a narrower version of several claims. However, these two issues are central to the paper's conceptual framing and broader significance, and they remain unresolved in my view. For this reason, I keep my score unchanged at Weak Reject (3).

**Key Questions For Authors:**

1. Regarding “self-knowledge”: In the experiment settings, the authors specify the model name in the prompt. However, is this sufficient for the model to recognize that the given “model name” refers to itself ?
2. I note that LoRA is only applied to the query and value matrices (Line 1040), and the learning rate is set to 1e-5, a value lower than the recommended range of [1e-4, 1e-3] in [1]. Is there a fundamental difference between the training of CM models and that of general tasks?
3. This paper applies a judge model to assess whether the response is correct or not to construct the training datasets. Since most of the conclusions are built upon the datasets, it is crucial to analyze the judgment error in this process.

[1]: https://thinkingmachines.ai/blog/lora/

**Limitations:**

yes

**Strengths And Weaknesses:**

Strengths
1. This paper introduces a novel perspective: the historical information of models plays an important role in model calibration.
2. Based on its theory, this paper trains Generalized Correctness Models using the historical information of different models, enabling a single LLM to model the correctness of many other models.
3. This paper investigates how the different components in the response affect the correctness predication performance.
4. This paper compares many history encoding methods, which are beneficial for improving calibration ability.

Weaknesses:
1. The experimental settings and results in this paper do not support the claim that “there is no privileged information for LLMs to predict their response correctness”.
- The method for LLMs to realize that the response is generated by them is to inject the model name into the prompt. Whether LLMs can understand the specified “model name” corresponding to them is still unknown. Thus, the results in Figure 2 or others can not justify the claim that the LLMs have little to no privileged information about their own correctness.
- The results in Figure 2 are based on fine-tuned LLMs. It remains unclear whether the phenomenon that different LLMs exhibit similar correctness prediction arises from the claim that LLMs have little privilege, as finetuning itself will make change to the model.
- For the non-tuned results in Table 8, we observe that when Qwen2.5-7B serves as both the correctness model and the target model, its AUROC is higher than that when Llama3.1-8B is used as the target model. The same trend holds for Llama3.1-8B when it acts as both the correctness and target model. These results demonstrate that LLMs are actually more proficient at predicting their own correctness confidence.
2. Is the model's historical information truly important? Since SCMs inject more model-specific historical information than GCMs, SCMs should outperform GCMs. However, this is inconsistent with the results in Table 2, Table 5, and Figure 3. The authors should conduct an additional experiment where the model names in the training data are shuffled or deleted to verify whether model history is critical for performance.
3. The baselines in this paper are relatively simple. Stronger calibration methods such as Temperature scaling and others should be included for comparison.
4. The results in this paper focus on the QA datasets. Whether the conclusions derived from such datasets can be generalized to other task types remains unknown.

---

> ### Author Rebuttal · Authors · 2026-03-31
>
> Thank you for highlighting that our paper introduces a ‘**novel perspective**’ that ‘**models have little to no privileged information about their own correctness**’ and ‘**compares many historical encoding methods**’.
>
> ### **W1. Further tests on untrained models to show little privileged information**
>
> You noted that fine tuning might prevent us from analyzing the claim whether LLMs have little privileged information, and observe some patterns in the AUROC numbers of our table covering the untrained condition that seem to suggest the existence of privileged information. To address this, we supplement with a grid of 25 comparisons in the untrained setting below. We show the AUROC in the table and observe that there is **no particular advantage in self prediction (the diagonal) vs cross prediction (the other entries)**. We will include results for all other metrics in the final paper (due to word limit).
> | Target \ Correctness   | Mistral-7B-Instruct   | Qwen2.5-7B-Instruct   | Llama-3.1-8B-Instruct   | Qwen3-8B   | gemma-3-4b-it   |
> |:-|:-|:-|:-|:-|:-|
> | Mistral-7B-Instruct    | 0.633      | 0.758       | 0.742      | 0.844      | 0.736    |
> | Qwen2.5-7B-Instruct    | 0.553        | 0.695  | 0.720   | 0.823     | 0.693       |
> | Llama-3.1-8B-Instruct  | 0.532         | 0.679 | 0.717   | 0.819      | 0.678       |
> | Qwen3-8B               | 0.506   | 0.710    | 0.716  | 0.794      | 0.633           |
> | gemma-3-4b-it          | 0.520    | 0.727    | 0.677     | 0.822      | 0.587      |
>
> We also note that we avoid making a blanket statement that there is no privileged information of any form and focus on privileged information for the purposes of obtaining a calibrated confidence score (lines 83-87). We’ll make this clearer in the final manuscript.
>
> ### **W2. Model Names and History**
> In conditions where the model has plenty of parametric knowledge, it may not be necessary to rely on names information, however, in scenarios where the model has little parametric information (e.g., something OOD from the model parametric training data), the model can rely only on historical information of performance in the past. We show that this gives a nontrivial prediction advantage (+20%) in Figure 3, the gap between answerless vs random.
>
> ### **W3 & W4. Request for Analyzing other Calibration Methods and Datasets**
> We conducted additional tests calibrating GCMs across 7 post-hoc calibration methods for TriviaQA and MMLU datasets, reporting averages across target 8 models (Included only TriviaQA below due to word count). We note that these calibration methods are **composable with the GCM** and can improve the GCM’s performance. We observe that for most methods, the improvement is small <1%, showing that **GCMs are relatively calibrated already** (usually less than 3% ECE) and might require hyperparameter tuning to benefit significantly from these methods. See also Table 26 of the appendix.
>
> Average TriviaQA Improvement
>
> | Method | ECE Improvement Mean ↑ | ECE Improvement SE | RMSCE Improvement Mean ↑ | RMSCE Improvement SE | N Models |
> |:-|:-:|:-:|:-:|:-:|:-:|
> | spline_calibration | +0.0023 | 0.0017 | +0.0021 | 0.0029 | 8 |
> | isotonic_regression | -0.0053 | 0.0037 | -0.0260 | 0.0056 | 8 |
> | temperature_scaling | +0.0027 | 0.0023 | +0.0022 | 0.0016 | 8 |
> | platt | -0.0134 | 0.0023 | -0.0214 | 0.0029 | 8 |
> | beta_calibration | +0.0009 | 0.0020 | +0.0001 | 0.0029 | 8 |
> | enir (ensemble near isotonic regression) | -0.0118 | 0.0026 | -0.0383 | 0.0038 | 8 |
> | bbq (bayesian binning by quartiles) | -0.0007 | 0.0022 | -0.0256 | 0.0083 | 8 |
>
> Other datasets: due to limited space, we refer to our reply to reviewer bJZi for **results on the Spider SQL program writing task [2].**
>
> ### **Questions**
>
> Self-knowledge: We note that previous works have analyzed self knowledge even without providing model name [1].
>
> Learning rate: We utilized the default LoRA target matrices from huggingface transformers and did not extensively tune the learning rate as we focused on analysis. We do not think there is a fundamental difference between training GCMs and other models. We will plan to supplement with an experiment training GCMs/SCMs with a different learning rate in the final paper.
>
> Judge decisions: We manually verify 30 judge decisions (MMLU split) and find 100% judgement accuracy, we’ll conduct a larger scale evaluation and include results in the final manuscript.
>
> Thank you for raising these important questions, and we hope our response will allow you to revisit your score.
>
> [1] Kadavath, Saurav, et al. "Language models (mostly) know what they know." arXiv preprint arXiv:2207.05221 (2022).
>
> [2] Yu, Tao, et al. "Spider: A large-scale human-labeled dataset for complex and cross-domain semantic parsing and text-to-sql task." Emnlp, 2018.

---

> > ### Author Rebuttal · Reviewer_oJfq · 2026-04-03
> >
> > Thank you for your detailed response, which has addressed some of our concerns. However, we still have reservations on several points as follows.
> >
> > 1. We note that the paper explicitly states its training and evaluation of confidence estimation are primarily conducted via logit-based confidence (P(True)). To avoid potential overclaiming, we recommend:  i) the paper’s core conclusions be explicitly bounded to this specific experimental setting; or ii) generalized to other calibration methods.
> > 2. We still have concerns regarding the rationale that inserting the model name into the prompt can "trigger" privileged self-information. At present, there is no evidence in this paper that the model exhibits differentiated behavior to different model names in the same response. We thus recommend that the authors supplement some behavioral evidence (e.g., demonstrating that the same response is assigned higher confidence when the prompt includes the target model’s own name versus other model names).
> > 3. To rigorously validate the core claim that "Confidence estimation is a generalizable, cross-model skill learned by systematically encoding correctness history rather than a model-specific introspection skill", we note that current experiments are restricted to datasets (MMLU and TriviaQA) that the target model handles proficiently. The validity of this claim in genuinely out-of-distribution (OOD) domains remains untested. We therefore recommend that the authors fully train and evaluate GCMs in OOD domains, and supplement this with a calibration performance comparison between the GCM and the model trained on OOD.

---

> > > ### Author Response · Authors · 2026-04-06
> > >
> > > ### **W1) Addressing Focus on Logit Based Confidences**
> > >
> > > We will make it clear that our core conclusions and focus is based on the logit confidence and training based domain. **While we still believe that the training based logit only domain is the most relevant domain for correctness prediction, we agree that it will be helpful to explicitly emphasize the setting more.**
> > >
> > > **Explanation for why we focused on logit based confidences**: One reason we picked logit based confidences is to follow prior work examining the issue of self knowledge [1]. We also tested 4 methods based on verbalized confidences in Table 2 and Table 7, and observe that these are outperformed by trained logit based methods.  In particular, verbalized methods suffer in AUROC due to coarse-grainedness; we find this is the case even if verbalized methods use much larger correctness models and also use RAG retrieval against the same historical data. Most other methods falling under uncertainty quantification are inference time only and also cannot be trained. We believe that logit confidences have the unique advantage of being easily trainable, whereas for example verbalized confidences would require RL with rollouts with sequential decoding to improve (much slower), or the existence of soft labels (also estimated by additional rollouts per example, or unobtainable in cases where labeling is non trivial).
> > >
> > > We will make our focus clear in the introduction for the final manuscript, especially following the claim in line 83, where we introduce the question “Are LLMs better than other LLMs at predicting their own correctness?”, clarifying that we analyze the logit based P(True) confidences.
> > >
> > > ### **W2) Model Names Inclusion**
> > >
> > > We do not seek to make any claim that simply including the name of the generating model is enough to trigger the knowledge without any training. We add the information of the name of the model only as an aid to training, so that the model can learn patterns of the performance of certain models on certain types of questions.
> > >
> > > Exp 1)
> > > We train a model with no model name info vs model with model name info trained in the full answerful condition on Qwen3-8B and Llama3.1-8B-Instruct. At test time we evaluate in the answerless condition. We show that by removing model names, information is lost (decrease of 5 points in AUROC), although other metrics show smaller decreases. This suggests even when training on strongly ID datasets (where parametric knowledge of correctness model is rich) and model name may not be relevant, the information is implicitly learned. However, we do recognize that in our current setup, the relevance of the model name information is less important than compared to other aspects of which we ablate from GCMs, we believe this is partially due to the correlation of questions that different models answer correctly in MMLU (Average Pairwise Correlation (Pearson) of 0.8520 across the 8 models we consider).
> > >
> > > | Condition | Accuracy | ECE | RMSCE | **AUROC** |
> > > |:----------|---------:|----:|------:|------:|
> > > | with_model_name | **0.704643** | **0.280859** | **0.188007** | **0.629063** |
> > > | without_model_name | 0.704358 | 0.290919 | 0.188861 | 0.572031 |
> > >
> > > ### **W3) Addressing Generalization Claims**
> > >
> > > We note that the claim we are making is that correctness prediction is a cross-model skill for the correctness model, not the target model. We show transfer for the skill of correctness prediction on one dataset (TriviaQA), to a more out of distribution dataset (Spider text to SQL task).
> > >
> > > Exp 2)
> > > We show a new experiment involving the **generalization of a GCM trained on TriviaQA to the Spider dataset**, in order to include something more out of distribution. In these cases, the fact that the performance is higher for the GCM vs the base model’s logits shows that the correctness prediction ability is transferable to another dataset (“generalizable, cross-model skill”). **We also do not seek to make any claim about strong dataset transferability, and ‘generalizable, cross-model skill’, refers to model transferability, which we showed in the main tables (Table 2,7).** We will make this clearer in the final manuscript and currently address this point in **lines (96-89) when introducing RQ2A.**
> > >
> > > | Target Model | Evaluated Model | Accuracy | ECE | RMSCE | AUROC |
> > > |:------|:----------------|---------:|----:|------:|------:|
> > > | Llama3.1-8B (Spider) | Base | 0.595429 | 0.350409 | 0.346986 | 0.871511 |
> > > | Llama3.1-8B (Spider) | GCM | **0.751429** | **0.078804** | **0.146247** | **0.904345** |
> > > | Gemma-3-27B (Spider) | Base | 0.587429 | 0.380419 | 0.418806 | 0.720603 |
> > > | Gemma-3-27B (Spider) | GCM | **0.711429** | **0.118741** | **0.176532** | **0.796972** |
> > >
> > > Thanks for your detailed reviews and followups, and we hope this response addresses your remaining questions and would allow you to improve your score!
> > >
> > > [1] Kadavath, Saurav, et al. "Language models (mostly) know what they know." arXiv preprint arXiv:2207.05221 (2022).

---

### Official Review · Reviewer_bJZi · 2026-03-18

**Soundness:** 4
**Presentation:** 4
**Significance:** 3
**Originality:** 2
**Overall Recommendation:** 5
**Confidence:** 4

**Summary:**

This paper has two key contributions related to predicting the correctness of a model's output. First, it challenges the notion that models have privileged "self-knowledge" by showing that fine-tuning one model to predict the correctness of another model's output works just as well as having a model predict the correctness of its own outputs. Second, it creates Generalized Correctness Models (GCM), which are trained on the combination of predictions from many models. The GCM is able to predict in general the correctness of outputs. Experiments on MMLU show that GCMs outperform similar models trained specifically for one model & task, as well as other baselines, both in terms of accuracy (predicting correct/incorrect) and expected calibration error. GCMs can generalize quite well to outputs of models not seen during training; they are somewhat uncalibrated when generalizing to new datasets, but this can be patched with post hoc calibration.

**Compliance With Llm Reviewing Policy:**

Affirmed.

**Final Justification:**

The authors added additional experiments on an additional dataset (SPIDER) and also investigated training on multiple datasets jointly. These experiments further strengthen the paper. Overall I believe this is strong empirical work and would be of great practical value to the community.

**Key Questions For Authors:**

Q1: I am not convinced by the assertion in paragraph 2 of the introduction that "history" is the missing component. I think typical post hoc calibration techniques like Platt scaling can also be interpreted as leveraging "history" (predictions on a development dataset). The real innovation is the fact that GCMs are trained on historical data from *multiple* models (as pointed out in the right column of Page 2), and I think this should be emphasized instead.

Q2: If I understand correctly, none of the GCMs in this paper are trained on *both* MMLU and TriviaQA. Is this correct? Would training on multiple datasets help with the dataset transferability issue?

A couple suggestions:
* It may be worth defining the term post hoc calibration. Intuitively I understand what this means but one could argue that the GCM is also a type of "post hoc calibration" since it comes after the answering process.
* If you would like citations for the notion that models have privileged self-knowledge of their own behavior, this paper may be a useful citation https://arxiv.org/abs/2502.13329

**Limitations:**

yes

**Strengths And Weaknesses:**

Overall this is a well-written and sound paper, though not especially novel or exciting. The contributions of the paper are solid and it provides useful recommendations as to how to predict correctness (use a GCM with post hoc calibration and the best possible LLM backbone). I also think the finding that models do not have special self-knowledge is useful and helps demystify some other results published in the community recently. The experimental results and analysis are thorough and provide clean takeaways for people who want to predict model correctness.

One key weakness is that the paper focuses heavily on MMLU, with some experiments on transferability to TriviaQA. There are many other tasks for which predicting correctness might be important, such as classification tasks (automated data labeling) or LLM-as-a-judge settings. The paper could be stronger if it evaluated in more diverse domains.

Another weakness is that the paper's technical methodology is quite simple, essentially fine-tuning LLMs on a prediction task. The technical novelty of the approach is quite limited. On the other hand, I do think the experimental design and findings are interesting.

---

> ### Author Rebuttal · Authors · 2026-03-31
>
> We thank you for highlighting that our paper helps **demystify** recent results surrounding model self-knowledge, and “**provide clean takeaways for people who want to predict model correctness**”. We address your review below:
>
> ### **W1. Experiments on additional Semantic Parsing Domain (Spider, Text to SQL)**
>
> Thank you for pointing out the need for experiments on more domains. In order to show that our methods work beyond MMLU, and further beyond the knowledge retrieval / QA setting, we conduct tests on Spider [1]. In Spider, the model must construct SQL queries in response to questions and a database schema. **We observe the same patterns and show that GCMs outperforms SCMs on this task**.
>
> | | Accuracy | ECE | RMSCE | AUROC |
> |:-----|-----------:|------:|--------:|--------:|
> | Untrained Predicts Qwen2.5-7B-Instruct | 0.617 | 0.339 | 0.295 | 0.81 |
> | SCM Predicts Qwen2.5-7B-Instruct | 0.766 | **0.031** | **0.1** | 0.838 |
> | GCM Predicts Qwen2.5-7B-Instruct |      **0.805** | 0.033 |   0.102 |   **0.891** |
>
> | | Accuracy | ECE | RMSCE | AUROC |
> |:------|-----------:|------:|--------:|--------:|
> | Untrained Predict Llama-3.1-8B-Instruct | 0.595 | 0.352 | 0.304 | 0.891 |
> | SCM Predict Llama-3.1-8B-Instruct | 0.873 | 0.028 | **0.089** | 0.901 |
> | GCM Predict Llama-3.1-8B-Instruct |      **0.898** | **0.023** |   0.09  |   **0.929** |
>
> Exp Detail: The setup for the experiments are the same as for our table 1 experiments, except for that we train the GCM to predict 6 models (rather than 8), skipping the 70B scale levels and training on the 32B models for time. We use the train set of Spider under the exact match evaluation to construct the correctness dataset.
>
> We evaluate all 6 models for Spider, and can include the final table in the main paper or our next reply if you would like to see them (the observed trends are similar across models). We additionally point to a full set of table-1 tests for TriviaQA which we include in our appendix as Table 2. As well as full comparisons between 16 specific correctness models across TriviaQA and MMLU vs general correctness models in tables 22 and 23.
>
> ### **W2. Novelty of Technical Methodology**
> We would like to highlight that our main contribution is not on technical methodology, but on the analysis findings, thus we focused on building well designed experimental setups, which you have also highlighted.
>
> ### **Q1. Why highlight History as the missing component?**
>
> Thank you for arguing that post hoc calibration techniques like Platt scaling can also be interpreted as leveraging "history". This is in fact our position: in our paper, we frame post-hoc calibration under “Alternative methods of encoding history” (line 110), and as mapping “historical confidences to correctness”. We made this decision because it allows us to talk about all of our analysis under one umbrella, and highlights the fact that all of our methods use historical information in some form, even beating the logit or verbalized confidences of much larger models without historical information.
>
> ### **Q2. Training on both datasets (MMLU and TriviaQA)**
>
> We **train a new model on both MMLU and TriviaQA** to study whether or not the transfer issue between datasets can be mitigated with the strategy. We see that by training a joint GCM, there is much better transfer across datasets. **Acc and AUROC improved on both datasets** compared to GCMs trained on only one dataset, and the big calibration error spike is largely mitigated (see especially RMSCE).
>
> MMLU Results
>
> | Target Model  |   Accuracy |   ECE |   RMSCE |   AUROC |
> |:--|--:|--:|--:|--:|
> |MMLU GCM ↓||||
> | Llama-3.1-8B-Instruct  | 0.820 | 0.023 |   0.080 |   0.890 |
> | gemma-3-27b-it  | 0.836 | 0.029 |   0.085 |   0.865 |
> |Joint GCM ↓|||||
> | Llama-3.1-8B-Instruct | **0.827** | **0.022** | **0.078** | **0.898** |
> | gemma-3-27b-it   | **0.84**  | **0.015** |  **0.074** | **0.871** |
>
> TriviaQA Results
>
> | Target Model  | Accuracy | ECE | RMSCE | AUROC |
> |:--|--:|--:|--:|--:|
> |TriviaQA GCM ↓||||
> | Llama-3.1-8B-Instruct |  0.847 | **0.029** |  **0.090** |   0.905  |
> | gemma-3-27b-it     |   0.862 | **0.028** |   **0.074** |   0.881 |
> |Joint GCM ↓|||||
> | Llama-3.1-8B-Instruct |      **0.849** | 0.046 | 0.103 |   **0.91**  |
> | gemma-3-27b-it  | **0.868** | 0.029 |   0.084 | **0.892** |
>
>
> ### **Addressing Suggestions**
>
> Thank you for the suggestions, we will include both in the final draft:
>
> We will define the term ‘post-hoc calibration’ as you have suggested, by pointing to the specific statistical techniques meant by this type of calibration, and separating it from DNN training based techniques like GCMs. We will add this paper in our related works section.
>
> We hope our response has addressed all of your questions and will allow you to revisit your score.
>
> [1] Yu, Tao, et al. "Spider: A large-scale human-labeled dataset for complex and cross-domain semantic parsing and text-to-sql task." Emnlp, 2018.

---

> > ### Author Rebuttal · Reviewer_bJZi · 2026-04-04
> >
> > Thank you for your thorough response. I appreciate the inclusion of the additional SPIDER experiments as well as the MMLU + TriviaQA merged experiments. I will raise my score to 5.

---

> > > ### Author Response · Authors · 2026-04-07
> > >
> > > Thank you for your engagement and for increasing your score, we are glad that our responses addressed your comments.

---

### Decision · Program_Chairs · 2026-04-30

**Decision:**

Accept (regular)

**Comment:**

The authors demonstrate that an LLM attempting to predict its own answer correctness performs no better than an unrelated LLM performing the same task, suggesting that confidence estimation is not a model-specific introspective skill. To address this, they propose Generalized Correctness Model, which is fine-tuned on historical correctness data from multiple model families to learn cross-model patterns of correctness prediction. Through experiments primarily on MMLU and TriviaQA, with additional validation on the Spider text-to-SQL dataset, the authors show that GCMs outperform both self-generated confidence estimates and model-specific correctness models.

The reviewers acknowledged that the paper is well-written, with sound experimental design and thorough empirical analysis. Despite these strengths, several concerns were raised by the reviewers. Reviewer bJZi noted that the focus heavily on MMLU limits broader generalizability claims, suggesting evaluation on more diverse domains like classification tasks or LLM-as-a-judge settings. Reviewer oJfq argued that the experimental settings do not adequately support the central claim that models lack privileged self-knowledge and noted that baseline comparisons were relatively simple without stronger calibration methods. Reviewer uuTW expressed concerns about how RLHF training might affect the linguistic mechanisms that GCMs rely upon for correctness prediction. Reviewer sZtR questioned the practical value of the approach for real-world deployment scenarios beyond benchmark tasks.

The authors provided the rebuttal with extensive results. They conducted new experiments on the Spider text-to-SQL dataset, demonstrating that GCMs maintain their advantage over SCMs in this more applied setting. They also provided an expanded grid of 25 untrained model comparisons showing no systematic advantage for self-prediction over cross-prediction. They also tested GCMs on non-RLHF models and showed successful transfer, mitigating this concern. Most concerns have been addressed by the rebuttal. The remaining concerns are that the interpretation of model-name effects remained insufficiently supported and the generalization claims in truly out-of-domain settings were not fully validated.

The paper makes a valuable empirical contribution by challenging existing assumptions about model self-knowledge and providing a practical framework for improved confidence estimation across model families. We encourage the authors to incorporate the clarifications from their rebuttal into the final manuscript.